# Low-temperature and circadian signals are integrated by the sigma factor SIG5

Dora L. Cano-Ramirez[1,2,9], Paige E. Panter[3,9], Tokiaki Takemura[4], Tara Saskia de Fraine[3], Luíza Lane de Barros Dantas [3], Richard Dekeya [3], Thiago Barros-Galvão[3], Pirita Paajanen [3], Annalisa Bellandi[3,5], Tom Batstone[2], Bethan F. Manley [2,6], Kan Tanaka[4], Sousuke Imamura[4,7], Keara A. Franklin[2], Heather Knight[8] & Antony N. Dodd [3] ✉

Chloroplasts are a common feature of plant cells and aspects of their metabolism, including photosynthesis, are influenced by low-temperature conditions. Chloroplasts contain a small circular genome that encodes essential components of the photosynthetic apparatus and chloroplast transcription/translation machinery. Here, we show that in *Arabidopsis*, a nuclear-encoded sigma factor that controls chloroplast transcription (SIGMA FACTOR5) contributes to adaptation to low-temperature conditions. This process involves the regulation of SIGMA FACTOR5 expression in response to cold by the bZIP transcription factors ELONGATED HYPOCOTYL5 and ELONGATED HYPOCOTYL5 HOMOLOG. The response of this pathway to cold is gated by the circadian clock, and it enhances photosynthetic efficiency during long-term cold and freezing exposure. We identify a process that integrates low-temperature and circadian signals, and modulates the response of chloroplasts to low-temperature conditions.

Low temperatures cause widespread alterations in the physiology and development of plants. Plants use a variety of regulatory mechanisms to respond to low-temperature conditions and to prepare for freezing temperatures through the process of cold acclimation[1,2]. Chloroplasts are essential for plant productivity and require resilience to cold temperatures because this impacts photoprotection, plastid genome transcription, membrane composition, reactive oxygen species metabolism, translation and the magnitude of photosystem II (PSII) excitation pressure[3–10]. A suite of mechanisms underlie the short- and longer-term responses of chloroplasts to low-temperature conditions. These derive from both nuclear-encoded proteins that affect chloroplast function and direct responses to cold within chloroplasts. For example, the cold-induced, nuclear-encoded and plastid-localized protein COR15A has a key role in providing freezing tolerance[5,11].

COR15A is localized to the chloroplast stroma and is thought to stabilize chloroplast membranes in response to the molecular crowding that occurs during freezing-induced cellular dehydration[12–14]. Furthermore, the chloroplast-localized galactolipid galactosyltransferase SENSITIVE TO FREEZING2 becomes active in response to cytoplasmic acidification during freezing, remodelling the chloroplast outer envelope to increase freezing tolerance[4,15–17]. Within chloroplasts, low temperatures cause rapid and reversible photoinhibition[18], which is thought to protect the photosynthetic apparatus from decreased biochemical activity in the presence of cold, including reduced rates of PSII repair[19]. Furthermore, moderate temperature reductions alter chloroplast ribosome occupancy, increasing the translation of specific chloroplast genes[10].

The majority of chloroplast proteins are encoded by the nuclear genome, yet chloroplasts also harbour a small circular genome that

[1]The Sainsbury Laboratory, University of Cambridge, Cambridge, UK. [2]School of Biological Sciences, University of Bristol, Bristol, UK. [3]John Innes Centre, Norwich, UK. [4]Laboratory for Chemistry and Life Science, Institute for Innovative Research, Tokyo Institute of Technology, Yokohama, Japan. [5]Laboratoire de Reproduction et Développement des Plantes, ENS de Lyon, Université de Lyon, UCBL, INRAE, CNRS, Lyon, France. [6]Wellcome Trust Sanger Institute, Hinxton, UK. [7]Space Environment and Energy Laboratories, Nippon Telegraph and Telephone Corporation, Musashino-shi, Japan. [8]Department of Biosciences, Durham University, Durham, UK. [9]These authors contributed equally: Dora L. Cano-Ramirez, Paige E. Panter. ✉e-mail: antony.dodd@jic.ac.uk

encodes essential components of the photosynthetic apparatus and chloroplast gene expression machinery. Chloroplast-encoded genes are transcribed by two RNA polymerases: plastid-encoded plastid RNA polymerase (PEP) and nuclear-encoded plastid RNA polymerase. PEP is a bacteria-like multi-subunit RNA polymerase that requires a σ70-like sigma factor for promoter recognition and transcription initiation[20–22]. Sigma factors are thought to have transferred from the plastid genome to the nuclear genome during the evolutionary history of plants, thus providing a mechanism for nuclear control of plastid transcription[20,21]. The *Arabidopsis thaliana* (*Arabidopsis*) nuclear genome encodes six sigma factors (SIGMA FACTOR1 (SIG1) to SIG6) that control chloroplast transcription during chloroplast biogenesis and steady-state photosynthesis[20,23–25]. The nuclear encoding of plastid sigma factors is thought to provide a set of signalling pathways from the nucleus to plastids[20,23]. For example, the sigma factor SIG5 participates in chloroplast transcriptional responses to light conditions[23,25,26], a variety of abiotic stresses[24,27] and the circadian regulation of specific chloroplast transcripts[24]. Here, we identified a new role for SIG5 in the responses of plants to low-temperature conditions.

## Results

### SIG5 communicates low-temperature information to chloroplasts

We investigated the hypothesis that sigma factors participate in low-temperature responses of chloroplasts, because transcripts encoding the *Arabidopsis* sigma factors *SIG1*, *SIG4* and *SIG5* accumulate in response to low temperatures (Extended Data Fig. 1)[23,28]. We focused on the role of SIG5 in low-temperature responses because published microarray data indicate that it has the greatest transcriptional response to cold (Extended Data Fig. 1)[28]. We used cold treatments of 4 °C for 3 h because this was the shortest cold treatment that provided a robust response of *SIG5* transcripts (Extended Data Fig. 1)[28]. We confirmed this using a quantitative polymerase chain reaction with reverse transcription (RT–qPCR; Fig. 1a, Extended Data Fig. 2a and Supplementary Data 1). ELONGATED HYPOCOTYL5 (HY5) is necessary for *SIG5* transcript accumulation in the light[23,29], and HY5 protein accumulates under low-temperature conditions owing to nuclear depletion of the ubiquitin ligase COP1 that targets HY5 for degradation[30]. This motivated us to investigate whether HY5 and HY5 HOMOLOG (HYH) contribute to the *SIG5* transcript response to low temperatures. When the cold treatment was given 1 h after dawn, we found that *SIG5* transcripts accumulate in response to 3 h of low temperatures in the wild type, but not in a *hy5 hyh* double mutant (Fig. 1b and Supplementary Data 1). Both *hy5* and *hyh* single mutants did not affect the response of *SIG5* transcripts to cold treatment at this time of day (Fig. 1b). Under control temperature conditions, *SIG5* transcripts accumulate predominantly in the light[23–26,29,31]. However, in darkness, *SIG5* transcript levels increased in response to a 3 h cold treatment in the wild type, but not in the *hy5 hyh* double mutant (Fig. 1c and Supplementary Data 1). It is known that *SIG5* transcript accumulation in response to salinity involves HOMEOBOX-LEUCINE ZIPPER PROTEIN17 (ATHB17)[27]. We found that cold induction of *SIG5* in the light was not altered in *athb17* mutants (Extended Data Fig. 2b), suggesting that ATHB17 does not participate in this response to cold.

Within chloroplasts, SIG5 controls transcription from the blue light responsive promoter (BLRP) of the chloroplast *psbDC* operon[23] and several other chloroplast genes[24]. Therefore, *psbD* BLRP transcript accumulation represents an informative read-out of SIG5 activity in plastids. We found that *psbD* BLRP transcripts accumulate strongly in response to 5 h cold treatment of the wild type, but not the well-characterized *sig5* loss-of-function mutant *sig5-3* (refs. [23,24,26]) (Fig. 1d and Supplementary Data 1). This indicates that SIG5 is necessary for the upregulation of chloroplast *psbD* BLRP transcript levels by cold treatment. *psbD* BLRP transcript abundance decreased in the *hy5 hyh* mutant in response to a cold treatment starting

1 h after dawn, indicating that HY5 and HYH are required for *psbD* BLRP transcripts to accumulate in response to cold (Fig. 1e). In contrast to *SIG5* transcripts, *psbD* BLRP transcripts did not accumulate in response to 5 h of cold treatment in darkness (Fig. 1c,f and Supplementary Data 1). Therefore, SIG5 is necessary for a cold-induced increase in chloroplast *psbD* BLRP transcripts in the presence of light, and this process requires HY5 or HYH.

### Circadian regulation by SIG5 involves HY5 and HYH

HY5 contributes to the circadian regulation of some transcripts[32,33], and SIG5 participates in circadian signalling to chloroplasts[24]. Therefore, we hypothesized that HY5 or HYH might contribute to circadian signalling to chloroplasts by SIG5. We investigated this by cultivating seedlings for 11 days under cycles of 12 h light and 12 h darkness, and then transferring the seedlings to conditions of constant (white) light and temperature to monitor the free-running rhythm of transcript abundance. Under these control temperature conditions, we compared the circadian rhythms of *SIG5* and *psbD* BLRP transcript accumulation in the *hy5*, *hyh* and *hy5 hyh* mutants, and the wild type. *SIG5* transcript abundance increased during the subjective night to reach a peak around subjective dawn (Extended Data Fig. 3a and Supplementary Data 1), which is consistent with other studies conducted under constant white light of 90 μmol m$^{-2}$ s$^{-1}$ (ref. [24]). This contrasts to the dynamics of *SIG5* transcript abundance under monochromatic light, where it peaks later in the subjective day[26]. The peak transcript abundance of *SIG5* was reduced significantly at a subset of time points in the *hy5* or *hyh* single mutants, and at a greater number of time points in the *hy5 hyh* double mutant (Extended Data Fig. 3a). The peak abundance of *psbD* BLRP was reduced significantly at some time points in the *hy5* mutant and *hy5 hyh* double mutant, but not in the *hyh* single mutant (Extended Data Fig. 3b and Supplementary Data 1).

To evaluate further the contribution of HY5 and HYH to circadian rhythms of *SIG5* and *psbD* BLRP transcript accumulation, we compared the amplitude of these rhythms in *hy5*, *hyh*, *hy5 hyh* and the wild type, using MetaCycle circadian rhythm analysis software[34]. Under these control temperature conditions, the amplitude of the circadian rhythm of *SIG5* and *psbD* BLRP transcript accumulation was lower in the *hy5*, *hyh* and *hy5 hyh* mutants compared with the wild type (Extended Data Fig. 3a,b and Supplementary Data 1 and 2). The amplitude was reduced more in *hy5* compared with *hyh*, and was comparable between *hy5* and *hy5 hyh* (Extended Data Fig. 3a,b and Supplementary Data 1 and 2). Therefore, either HY5 and HYH participate in the circadian regulation of *SIG5* transcript accumulation, or alternatively HY5 or HYH allows another factor to confer the circadian rhythm of *SIG5* transcript accumulation. This does not completely explain the circadian control of *SIG5* transcript abundance, because transcript levels continue to oscillate with low amplitude in *hy5 hyh* (Extended Data Fig. 3a and Supplementary Data 1 and 2). We transiently expressed *SIG5::LUCIFERASE* in the wild type and *hy5*, *hyh* and *hyh5 hyh* mutants using particle bombardment[35,36] and found that *SIG5* promoter activity was reduced substantially in the *hy5* and *hy5 hyh* double mutants, and partially in the *hyh* mutant, relative to the wild type (Extended Data Fig. 3c). This supports the notion that HY5 and HYH are important regulators of SIG5 promoter activity[23,29,37]. *psbD* BLRP transcripts had a late phase or longer period in the *hy5* and *hy5 hyh* mutants, and were arrhythmic in *hyh* (Extended Data Fig. 3b and Supplementary Data 2; MetaCycle BH.Q, $P = 0.27$ for *hyh*), which differs from the rhythm of *SIG5* transcript accumulation (Extended Data Fig. 3a). One interpretation is that additional effects of *hy5* and *hyh* on this pathway, downstream of SIG5, contribute to the circadian regulation of *psbD* BLRP transcript levels. This difference is supported by HY5 and HYH also having differing roles in the responses to low temperatures of *SIG5* and *psbD* BLRP (Fig. 1b–f). *HY5* and *HYH* transcript levels were upregulated by a 3 h cold treatment at either ZT25 or ZT37 (Extended Data Fig. 3d), with the exception of *HYH* in the Ws background at ZT37.

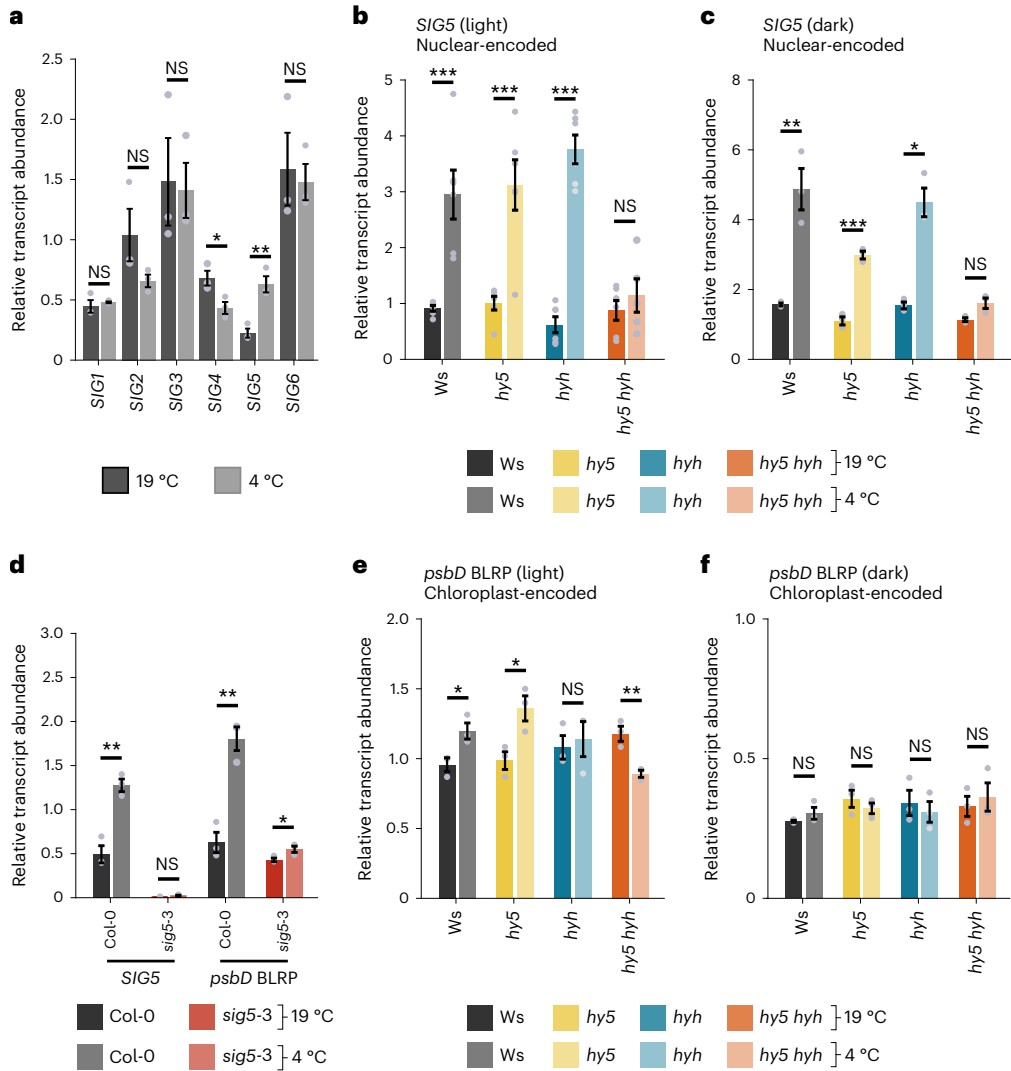

**Fig. 1 | SIG5 communicates information to chloroplasts about cold temperature conditions, and this requires HY5 and HYH. a**, Relative abundance of all six *Arabidopsis* sigma factor transcripts in wild type (Col-0) after 3 h at 19 or 4 °C. **b,c**, *SIG5* transcript accumulation in wild type (Ws), *hy5*, *hyh* and *hy5 hyh* double mutant after 3 h at 4 °C in light (**b**) and darkness (**c**). **d**, Abundance of *SIG5* and chloroplast *psbD* BLRP transcripts in Col-0 and *sig5*-3 mutant after 3 h (*SIG5*) and 5 h (*psbD* BLRP) at 4 °C. **e,f**, *psbD* BLRP transcript accumulation in Ws, *hy5*, *hyh* and *hy5 hyh* double mutant after 5 h at 4 °C in light (**e**) and darkness (**f**). Darker and paler bars indicate control (19 °C) and cold (4 °C)

treatments, respectively. Experiments used 11-day-old seedlings. *SIG5* and *psbD* BLRP transcript abundance was measured after 3 and 5 h of cold treatment, respectively, because there is a time delay between the accumulation of *SIG5* transcripts and downstream *psbD* BLRP[24,26]. Data represent mean ± s.e.m. and *n* = 3, except in **b** where *n* = 6. Statistical significance represents cold treatments compared with control temperature conditions (two-sided *t*-tests). ***$P < 0.001$; **$P < 0.01$; *$P < 0.05$; NS, not significant. Exact *P* values are given in Supplementary Data 1.

## Circadian gating of the cold response of *SIG5*

We tested the hypothesis that there is circadian gating of the responses of *SIG5* and *psbD* BLRP transcripts to cold, because there is circadian gating of other transcriptional responses to low temperatures[38,39]. Circadian gating is the process whereby the circadian oscillator modulates the response to a stimulus, so that the magnitude of the response depends on the time of day of the stimulus[40,41].

Groups of seedlings were exposed to 3 h cold treatments, at regular intervals, under constant light conditions. Each separate group of seedlings received a single cold treatment and was then harvested to measure the response of the transcripts to a cold treatment given at that particular time. Cold treatment caused greater *SIG5* transcript accumulation between subjective midnight (zeitgeber time (ZT) 41; that is, 41 h after the final dawn under constant free-running conditions) and subjective dawn (ZT49), and less accumulation between

ZT33 and ZT37 (Fig. 2a and Supplementary Data 1). This suggests that there is circadian gating of the response of *SIG5* transcripts to cold. Cold caused greatest *psbD* BLRP transcript accumulation during the subjective day, compared with a peak at subjective dawn under control temperature conditions (Fig. 2b and Supplementary Data 1). The phase shift of *psbD* BLRP after short cold treatments (Supplementary Data 2) might suggest the cold-responsive circadian gate is timed with a different phase compared with the control-temperature circadian rhythm, or that low temperature delays *psbD* BLRP transcript accumulation. In the *sig5*-3 mutant, *psbD* BLRP remained cold-inducible at several time points during the subjective day (Fig. 2b, red symbols), suggesting that *psbD* BLRP transcript levels are regulated by a mechanism additional to SIG5. This pattern of circadian gating of cold induction of *SIG5* transcripts is altered at some time points in the *hy5* mutant (Fig. 2c and Supplementary Data 1), unaffected in *hyh* and abolished in the

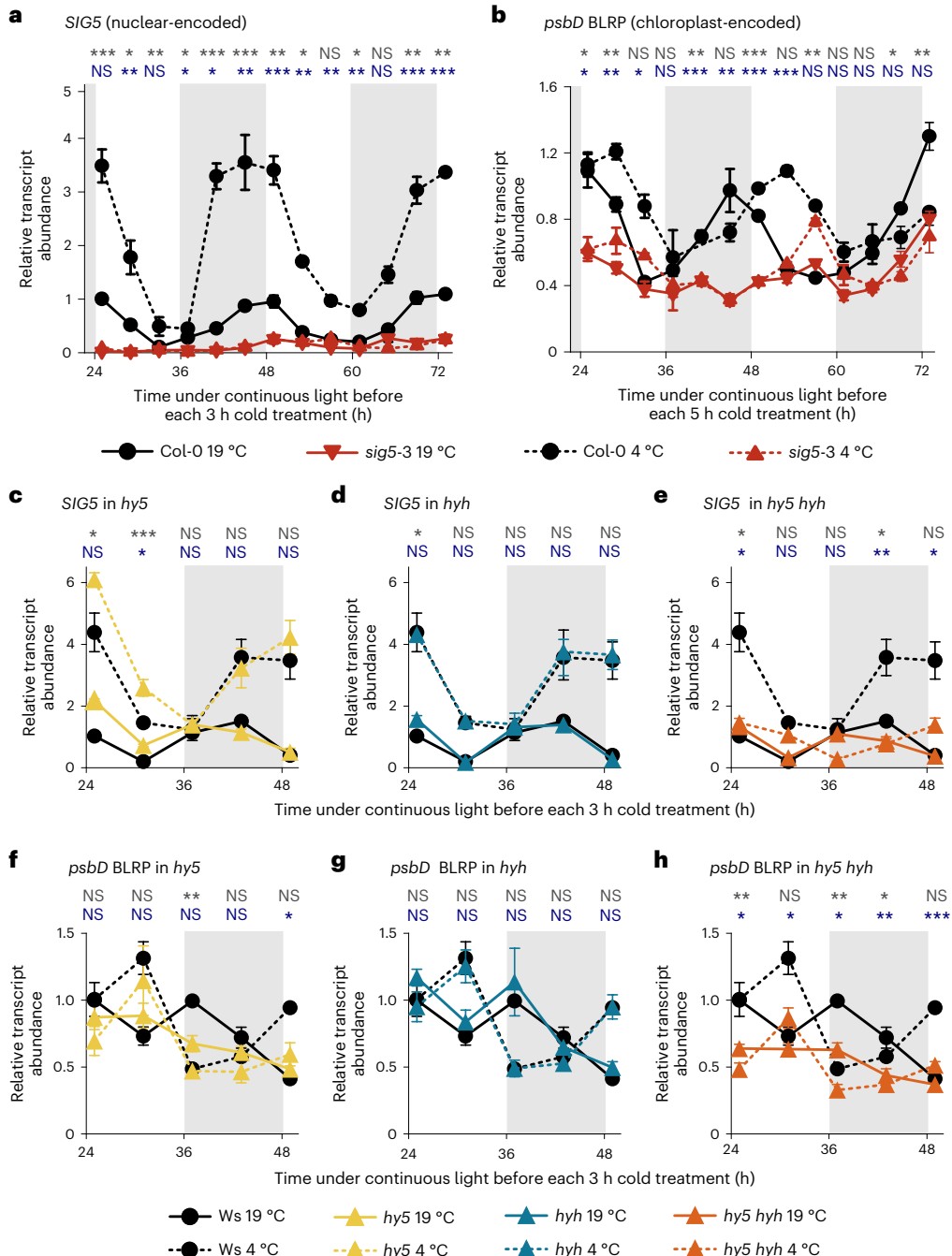

**Fig. 2 | Circadian gating of the responses to cold of *SIG5* and chloroplast *psbD* BLRP, and the involvement of HY5 and HYH. a,b,** Circadian gating of the response to cold of *SIG5* (**a**) and *psbD* BLRP (**b**) transcripts in the Col-0 wild type and *sig5*-3 mutant. **c–f,** Circadian gating of the response to cold of *SIG5* (**c–e**) and *psbD* BLRP (**f–h**) in the *hy5*, *hyh* or *hy5 hyh* double mutant. Cold treatments comprised 3 h at 4 °C for *SIG5* transcript levels, and 5 h at 4 °C for *psbD* BLRP. Each short cold treatment was applied to a separate batch of seedlings. The *x* axis indicates the time at which the cold treatment commenced. Grey shading on graphs indicates subjective night, under constant light conditions. Solid and broken lines indicate control (19 °C) and cold (4 °C) treatments, respectively. Wild-type data (black lines) are duplicated across **c–e** and **f–h** for visual clarity. Experiments used 11-day-old seedlings. Data represent mean ± s.e.m. of three independent biological replicates. Statistical information above graphs compares the transcript levels in the wild type and mutant under control temperature conditions (grey text) and in response to cold (blue text) at each time point. ***$P < 0.001$; **$P < 0.01$; *$P < 0.05$; NS, not significant in unpaired two-sided *t*-tests. Exact *P* values are given in Supplementary Data 1.

*hy5 hyh* double mutant (Fig. 2c–e, *hy5 hyh* in Fig. 2e and Supplementary Data 1). The circadian gating of cold induction of *psbD* BLRP is also altered at some time points in the *hy5* mutant and *hy5 hyh* double mutant (Fig. 2f–h). In general, *psbD* BLRP appears less cold-responsive in the Ws accession compared with Col-0 (Figs. 1d,e and 2b,f), which is consistent with differences in temperature responses between *Arabidopsis* accessions[42,43]. It appears that the *hy5* single mutant affects *SIG5* and *psbD* BLRP transcript accumulation at control temperatures, whereas the *hy5 hyh* mutant is required to abolish its response to cold (Extended Data Fig. 2a,b and Fig. 2f–h).

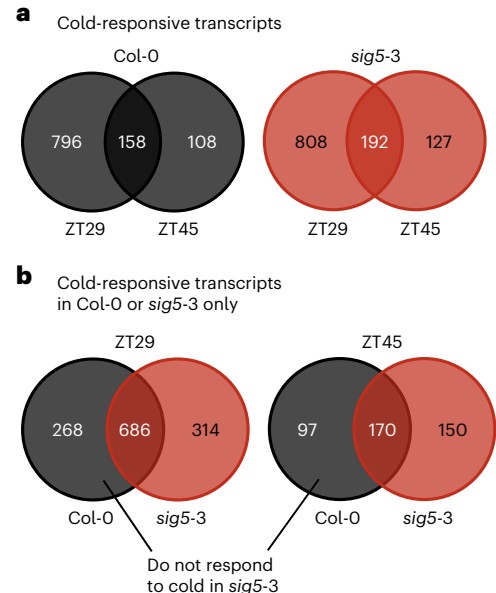

**a** Cold-responsive transcripts

Col-0                                                        *sig5*-3

796      158      108                  808      192      127

ZT29      ZT45                          ZT29      ZT45

**b** Cold-responsive transcripts
in Col-0 or *sig5*-3 only

ZT29                                              ZT45

268      686      314                    97      170      150

Col-0      *sig5*-3                        Col-0      *sig5*-3

Do not respond
to cold in *sig5*-3

**Fig. 3 | Genome-wide influence of SIG5 upon the cold-responsive transcriptome. a**, Overlap between transcripts responsive to cold in Col-0 or *sig5*-3 at time points ZT29 and ZT45. **b**, Overlap between transcripts responsive to cold in Col-0 and *sig5*-3 at the two time points. Numbers within the circles on Venn diagrams indicate the number of transcripts. Experiments used 11-day-old seedlings and 3 h (ZT45) or 5 h (ZT29) cold (4 °C) treatments. ZT refers to the time elapsed under free-running (constant) conditions, after the final dawn.

## SIG5 shapes the nuclear-encoded cold-responsive transcriptome

We hypothesized that SIG5 might have a broader role in cold-responsive gene regulation, because other sigma factors can indirectly influence nuclear-encoded gene expression[44,45]. To test this, we investigated transcriptome alterations in wild-type and *sig5*-3 seedlings in response to 3 h cold treatments given at two different time points (ZT29 and ZT45). We selected these times because they correspond to the peak sensitivity of *psbD* BLRP and *SIG5* to cold, respectively (Fig. 2a,b). Under control temperature conditions, a relatively small number of transcripts was differentially expressed between Col-0 and *sig5*-3 at the time points examined (33 and 42 transcripts were differentially expressed at ZT29 and ZT45, respectively), with no significant Gene Ontology (GO)-term enrichments within these gene sets (Supplementary Data 3). In the Col-0 wild type, 954 and 266 transcripts responded to cold at ZT29 and ZT45, respectively, whereas in *sig5*-3, 1,000 (ZT29) and 319 (ZT45) transcripts responded to cold (Fig. 3a and Supplementary Data 3 and 4; cold-responsive defined as log(fold change) > 2 and *P* ≤ 0.01 using Voom/Limma method[46]). Some 158 transcripts in Col-0 (13% of cold-responsive transcripts) and 192 transcripts (14.6%) in *sig5*-3 responded to cold at both time points, so the majority of cold-responsive transcripts were unique to the time at which the seedlings were cold-treated (Fig. 3a). The different sets of cold-responsive transcripts at the two time points are consistent with the notion that there is circadian gating of the cold-responsive transcriptome in plants[47]. We compared our cold-responsive transcript set in Col-0 with that of Zhao et al. (3 h cold treatment)[48] and found that 42.5% (ZT29) and 13.4% (ZT45) of cold-responsive transcripts were shared between the studies (Extended Data Fig. 4a)[29]. The smaller overlap at ZT45 might reflect time of day differences in the cold-responsive transcriptome.

Of the transcripts that responded to cold at ZT29, 268 were cold-responsive only in Col-0 (198 upregulated, 70 downregulated) but not cold-responsive in *sig5*-3 (Fig. 3b and Supplementary Data 5; statistical threshold for cold-responsiveness of log(fold change) > 2 and

*P* ≤ 0.01 using Voom/Limma method[46]). Similarly, at ZT45, 97 transcripts responded to cold (46 upregulated, 51 downregulated) in Col-0 but not *sig5*-3 (Fig. 3b and Supplementary Data 5). At ZT29, 314 transcripts responded to cold in *sig5*-3 but not Col-0 (179 upregulated, 135 downregulated), whereas at ZT45 150 transcripts responded to cold in *sig5*-3 only (109 upregulated, 41 downregulated, including several chloroplast transcripts; Fig. 3b and Supplementary Data 5). Together, this indicates that some transcripts required SIG5 to respond significantly to the cold treatment. Because circadian timing influences the response to cold of *psbD* BLRP transcripts (Fig. 2b), we hypothesized that the set of transcripts that responded significantly to the cold treatment in the wild type but not *sig5*-3 mutant might be enriched with circadian-regulated transcripts. However, examination of cold-responsive transcript sets that are unique to Col-0 or *sig5*-3 identified that circadian-regulated transcripts[49,50] were not overrepresented among the *sig5*-3-specific cold-responsive transcripts. Furthermore, the set of transcripts that responded significantly to the cold treatment in Col-0 but not in *sig5*-3 was significantly underrepresented with circadian-regulated genes (Extended Data Fig. 4b). The only circadian clock-associated transcript that was significantly cold-induced in Col-0 but not *sig5*-3 was *NIGHT LIGHT-INDUCIBLE AND CLOCK-REGULATED4* (*LNK4*) (at ZT29; Supplementary Data 5), although the role of LNK4 within circadian regulation remains somewhat uncertain[51,52]. Furthermore, transcripts encoding the zinc-finger protein B-BOX DOMAIN PROTEIN19 (BBX19) were upregulated by cold in *sig5*-3, but not the wild type (Supplementary Data 5). BBX19 is thought to repress the promoters of certain morning-phased circadian clock components[53]. There was a significant overlap between transcripts that responded significantly to the cold treatment in *sig5*-3 but not in the wild type and putative HY5 targets[37] (Extended Data Fig. 4c), but no significant intersection with HY5 regulated cold-induced genes (Extended Data Fig. 4d)[30].

Using GO-term analysis, we evaluated whether the sets of transcripts that responded significantly to cold in only Col-0 or *sig5*-3 are enriched with genes linked to specific processes. Genes linked to hypoxia responses were overrepresented at ZT29 for transcripts that responded significantly to cold in Col-0 but not *sig5*-3 (Benjamini–Hochberg correction, *P* < 0.05), and in a combined list of transcripts that responded significantly to cold in only *sig5*-3 at both ZT29 and ZT45 (*P* < 0.05) (Supplementary Data 6). The set of cold-responsive transcripts in *sig5*-3 only was enriched for AP2/ERF domain proteins (*P* < 0.001), which participate in abiotic and biotic stress responses, growth and development[54].

## SIG5 maintains photosynthetic efficiency during low-temperature conditions

We reasoned that the cold induction of transcripts encoding *SIG5* and its chloroplast target *psbD* BLRP might underlie physiological responses of plants to low temperatures. We investigated the involvement of SIG5 in cold and freezing responses using chlorophyll fluorescence as a proxy for photosynthetic responses to cold and freezing, and electrolyte leakage as a measure of tissue damage by freezing.

Because SIG5 regulates the transcription of the gene encoding the D2 protein of PSII (*psbD*)[23] and low temperature increases PSII excitation pressure[55], we investigated PSII photosynthetic efficiency by measuring chlorophyll fluorescence ($F_v/F_m$) in *sig5*-3 after short- and long-term cold treatments. In the wild type, cold reduced the ratio of variable fluorescence ($F_v$) to maximum fluorescence ($F_m$), $F_v/F_m$ relative to the 20 °C control (Fig. 4a). Furthermore, $F_v/F_m$ was reduced significantly in *sig5*-3 compared with the wild type after a long-term cold treatment of 10 days at 4 °C (Fig. 4a). A short freezing treatment of cold-acclimated plants (−8 °C for 6 h) decreased $F_v/F_m$, with $F_v/F_m$ in *sig5*-3 reduced significantly more than in the wild type (Fig. 4a,b). Therefore, SIG5 contributes to maintaining the photosynthetic efficiency of PSII during prolonged cold and short-term freezing. We reasoned that this might arise from effects of the *sig5*-3 mutation upon photosystem

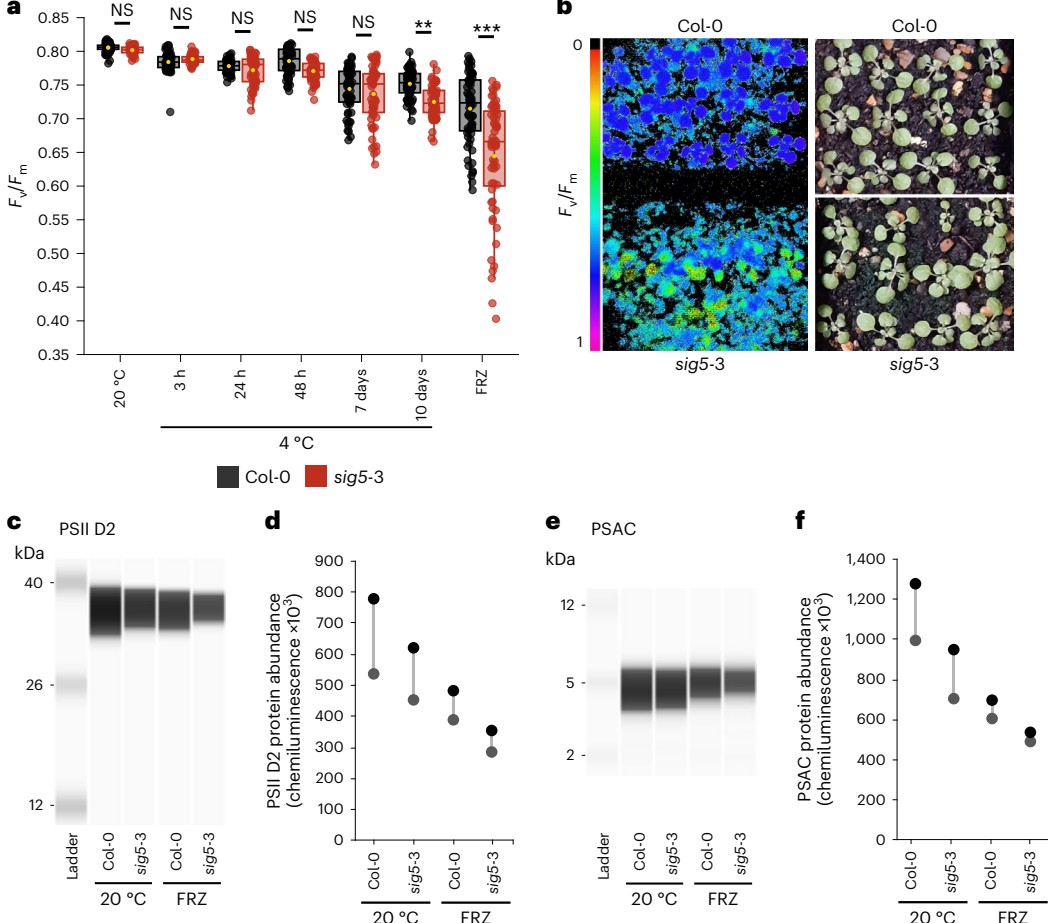

**Fig. 4 | SIG5 influences photosynthetic efficiency under cold temperature conditions. a**, Photosynthetic efficiency of PSII ($F_v/F_m$) of 14-day-old wild-type (Col-0) and *sig5*-3 plants exposed to cold (4 °C) and freezing (FRZ) treatments ($n = 60$). In box plots, the box indicates the interquartile zone with the median line at the centre, whiskers indicate interquartile range and a yellow dot indicates the mean. Data were analysed by two-way analysis of variance followed by post-hoc Tukey test. ***$P < 0.001$; **$P < 0.01$; NS, not significant ($P$ values: 20 °C = 0.999; 3 h = 0.999; 24 h = 0.999; 48 h = 0.474; 7 days = 0.994; 10 days = 0.001; FRZ < 0.001). **b**, $F_v/F_m$ of Col-0 and *sig5*-3 seedlings after freezing at −8 °C for 6 h (left) and representative image of these plants before freezing (right).

**c**–**f**, Automated semiquantitative capillary immunoassay comparing PSII D2 (**c**,**d**) and PSAC protein (**e**,**f**) levels between wild type (Col-0) and *sig5*-3 under control temperature conditions (20 °C) and after freezing at −8 °C for 6 h (FRZ). Samples were analysed in triplicate (three technical replicates) from each independent experiment, with (**d**,**f**) two independent experiments shown. **d**,**f**, Quantification of PSII D2 (**d**) and PSAC protein (**f**) levels (area under each peak), relative to levels in wild type (Col-0) at 20 °C from replicate immunoassays. Circles on plots indicate result from each independent experiment. Cold (4 °C) and freezing treatments (FRZ) were conducted identically for all experiments and included a 10-day cold acclimation period at 4 °C before the freezing treatment.

protein abundance during freezing. To investigate this, we compared after the short freezing treatment that reduced $F_v/F_m$, the abundance of a photosystem protein that is regulated transcriptionally by SIG5 (PSII D2) and a protein thought to not be regulated transcriptionally by SIG5 (PSAC)[24]. Both PSII D2 and PSAC protein abundance was decreased consistently in *sig5*-3 plants after this freezing treatment, compared with Col-0 under control temperature conditions (Fig. 4c–f and Extended Data Fig. 5a,b). By contrast, a chloroplast-encoded and localized protein that does not form part of the photosystems (RbcL), which is not thought to be regulated by SIG5 (ref. 24), was unaltered in *sig5*-3 by this freezing treatment (Extended Data Fig. 6a–c). Normalization of PSII D2 and PSAC protein abundance to the abundance of RbcL under each treatment confirms the reduced abundance of these photosystem proteins relative to RbcL (Extended Data Fig. 6c,d). This suggests that reduced PSII D2 abundance might occur through either a direct effect of the *sig5*-3 mutation upon *psbD* BLRP promoter activity, or alternatively through a general alteration in photosystem protein levels in *sig5*-3 after freezing. *SIG5* and *psbD* BLRP transcript levels were decreased relative to control temperature conditions after freezing (Extended Data Fig. 7a), suggesting the presence of SIG5 rather than its cold induction

maintains PSII D2 and PSAC protein abundance during freezing. $F_v/F_m$ was also reduced significantly in *hy5 hyh* mutants compared with the wild type (Ws) after 10 days at 4 °C, and after 6 h of freezing (Extended Data Fig. 7b), so regulation by HY5 and/or HYH also maintains PSII photosynthetic efficiency at low temperatures.

During cold acclimation, sensing of low non-freezing temperatures leads to changes in membrane fluidity, cell wall structure and the accumulation of compatible solutes and antioxidants to increase freezing tolerance[1,5,56]. Previous studies have shown that greater damage to the photosynthetic apparatus of plants with lower freezing tolerance can manifest as reduced $F_v/F_m$ during freezing[57], as we identified for *sig5*-3 (Fig. 4a). For example, reduced $F_v/F_m$ of *hy5 hyh* after prolonged cold and freezing (Extended Data Fig. 7b) is consistent with reduced freezing tolerance of the *hy5* mutant, compared with the wild type, after cold acclimation[19]. We tested whether there was reduced freezing tolerance in *sig5*-3, but found no difference in the survival of 14-day-old *sig5*-3 and wild-type plants that were cold-acclimated for 10 days at 4 °C and then subjected to −8 °C (Extended Data Fig. 7c,d). We also tested this in mature rosette plants, using the freezing-induced leakage of electrolytes from leaf discs from rosette leaves as a proxy

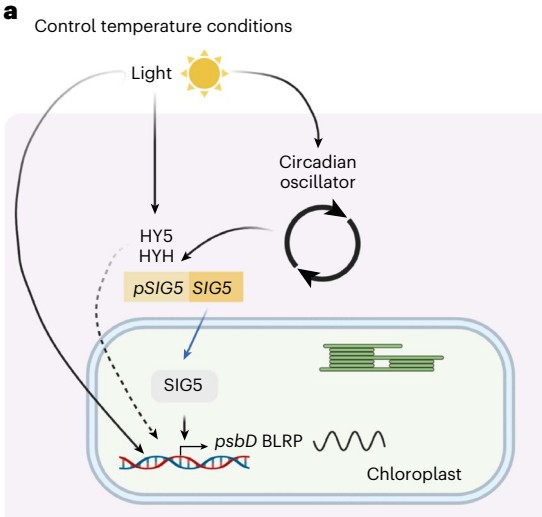

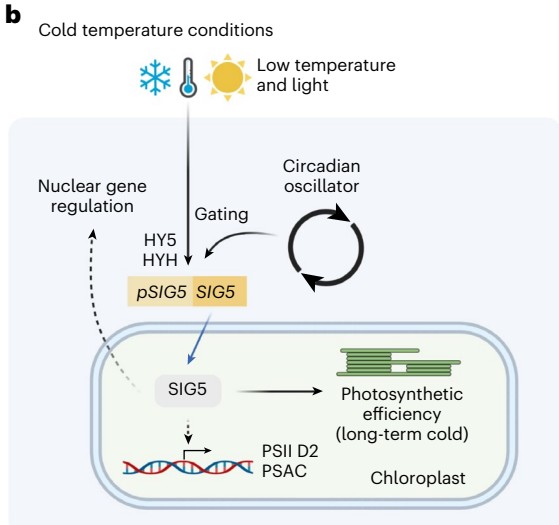

**Fig. 5 | Involvement of SIG5 in cold-temperature responses. a**, Under control temperature conditions, light regulates the circadian clock and also HY5 and HYH-regulated genes. HY5 and HYH are necessary for the circadian regulation of *SIG5* transcript accumulation, and circadian clock components might also regulate *SIG5* expression directly. SIG5 regulates transcription of *psbD* via the BLRP, and HY5 or HYH might regulate *psbD* BLRP transcription through additional mechanisms. **b**, In response to cold temperatures, HY5 and HYH are necessary for the accumulation of *SIG5* transcripts in response to cold, and the circadian clock gates the response to cold of *SIG5* transcripts. SIG5

regulates PSII D2 and PSAC protein abundance, either by direct transcriptional regulation or through indirect mechanisms. SIG5 is necessary to maintain photosynthetic efficiency under long-term cold. SIG5 mutants have altered nuclear gene expression in response to cold, suggesting that SIG5 indirectly regulates nuclear genome transcription. Black solid arrows indicate regulatory relationships, broken arrows indicate inferred connections, a blue arrow entering the chloroplast indicates SIG5 targeting to chloroplasts and the sine wave icon indicates circadian regulation. Subcellular localization is inferred.

for freezing damage. This indicated that there was no difference in the level of cellular damage between *sig5*-3 and wild type, irrespective of cold acclimation (Extended Data Fig. 7e,f), even though $F_v/F_m$ was also reduced after 14 days at 4 °C in leaves of mature rosettes (Extended Data Fig. 7g). We conducted electrolyte leakage analysis on mature leaves rather than younger plants to enable the very consistent sampling of leaf discs that is necessary to limit data noise. Nevertheless, freezing tolerance was unaltered by the *sig5*-3 mutant, relative to the wild type, in both 14-day-old seedlings and mature plants (Extended Data Fig. 7c–f). Therefore, basal or acquired freezing tolerance under the conditions tested is unaltered in *sig5*-3, and the lower photosynthetic efficiency of the mutant during long-term cold (Fig. 4a) does not affect freezing survival (Extended Data Fig. 7c–f).

## Discussion

We found that SIG5 is required for a response of chloroplast-encoded *psbD* BLRP transcripts to cold, suggesting that SIG5 communicates information to chloroplasts about low-temperature conditions. This response involves HY5 and HYH (Fig. 5a,b). HY5 also contributes to chloroplast processes such as photopigment biosynthesis[33,58], so probably influences chloroplast gene expression through multiple independent mechanisms. HY5 is necessary for *SIG5* transcript accumulation in response to light[23,29], so regulation of SIG5 by HY5 and HYH integrates several environmental cues that are communicated to chloroplasts.

We found that in darkness, low-temperature induction of nuclear-encoded SIG5 was not accompanied by upregulation of *psbD* BLRP (Fig. 1c,f). One interpretation is that in darkness, low temperature upregulates *SIG5* transcript levels, which might increase SIG5 protein abundance within chloroplasts. However, light is required for the association of PEP with chloroplast DNA, for PEP assembly and regulation of sigma factor phosphorylation state, with possible involvement of redox regulation[59–63]. Therefore, upregulation of *SIG5* by cold in darkness might not alter *psbD* BLRP transcription because PEP is inactive. Another possibility is that in darkness, SIG5 is not imported efficiently into chloroplasts, so does not reach a threshold required to

generate *psbD* BLRP transcripts[64]. Overexpression of *SIG5* to a very high level from the chloroplast genome of transplastomic plants constitutively upregulates *psbD* BLRP, even in darkness[65]. This difference from our results might reflect the very high expression levels that are possible in transplastomic plants[65]. Either way, these interpretations support the notion that additional regulatory steps are positioned between *SIG5* expression and accumulation of *psbD* BLRP transcripts.

There was a difference in the circadian phase of cold-sensitivity of *SIG5* and *psbD* BLRP transcripts, such that *SIG5* had greatest cold-responsiveness towards the end of the subjective night, whereas *psbD* BLRP transcripts had greatest cold-responsiveness during the subjective day (Fig. 2a,b). There are several potential explanations for this timing difference. The process of SIG5 protein synthesis, chloroplast import and PEP assembly will take some time, introducing a time delay into the process. Such a delay between *SIG5* and *psbD* BLRP transcript accumulation also occurs under light/dark cycles at control temperatures[24], and in field-grown plants[66]. The circadian clock might also influence chloroplast protein import, the expression of PEP-associated proteins required for chloroplast transcription[67], and the activity of protein kinases thought to modulate sigma factor function such as redox-responsive CHLOROPLAST SENSOR KINASE[62,68]. In combination, these factors could superimpose several layers of temporal regulation upon this process of gene regulation.

One interpretation of our results is that in response to cold, SIG5 regulates *psbD* BLRP transcription directly to increase the supply of messenger RNA for translation into PSII D2, maintaining photosynthetic activity. Alternatively, SIG5 might regulate PSII D2 protein abundance independently from its role in *psbD* BLRP transcription, because chloroplast protein abundance is regulated by transcript stability, translation and protein turnover[69–71]. For example, other sigma factors regulate chloroplast transfer RNA expression[44,72,73] and a chloroplast-encoded subunit of the ATP-dependent ClpP (caseinolytic) protease[74], opening the possibility that SIG5 might influence PSII D2 accumulation through mechanisms such as translational regulation or protein degradation. The second interpretation appears to be

supported by our data, because *psaC* is not thought to be a target of SIG5 regulation[24], yet PSAC protein abundance decreases after freezing in *sig5*-3 (Fig. 4e,f). This suggests a more general role for SIG5 in maintaining photosystem protein abundance under certain stress conditions than thought previously.

SIG5 is required for circadian regulation of a set of chloroplast transcripts[26]. We identified that HY5 and HYH, acting with additional mechanisms, contribute to circadian rhythms of *SIG5* transcript accumulation under control temperature conditions (Extended Data Fig. 3a,b). Given that HY5 coregulates transcripts with PHYTOCHROME INTERACTING FACTORs (PIFs) and there are interactions between CCA1 and HY5 proteins for the regulation of promoter activity[32,33], circadian oscillator components or PIFs could contribute directly to the circadian regulation of *SIG5*. For example, chromatin immunoprecipitation experiments indicate that PIF1 and the circadian clock component PRR5 bind to the *SIG5* promoter[75,76], whereas CCA1 and LHY do not[77–79]. Therefore, multiple circadian clock-related factors appear to converge upon the promoter of SIG5, with HY5 and HYH representing one of these mechanisms. The circadian regulation and low-temperature responses of *SIG5* transcripts could occur through HY5 and HYH regulating the G-box motif within the *SIG5* promoter[29].

It is interesting that a set of nuclear-encoded transcripts are cold-responsive in the wild type, but not in *sig5*-3. This phenotype of the *sig5*-3 mutant indicates that a function of SIG5 can influence nuclear-encoded gene expression. We speculate that this probably occurs indirectly, perhaps through metabolic alterations arising from altered chloroplast function in the *sig5*-3 mutant.

## Conclusions and perspectives

Sigma factors allow bacteria and cyanobacteria to respond to cold temperature conditions[80–83]. Our experiments identify that sigma factors also participate in responses to cold temperatures in plants. Therefore, taken together with studies in bacteria and cyanobacteria[80–83], it appears that sigma factors are involved in cold-temperature responses in both prokaryotes and eukaryotes. Our experiments identify a new regulator of cold-temperature responses of chloroplasts, and establish that a sigma factor contributes to protection of photosynthesis before and during freezing. The greater cold-sensitivity of this signalling pathway immediately before subjective dawn, combined with its role in light stress responses[23], suggests it might be important during cold, bright mornings.

## Methods

### Plant material and growth conditions

All experiments were conducted using *Arabidopsis thaliana* L. (Heynh.). *Arabidopsis* seeds were surface-sterilized[26] and sown on half strength Murashige and Skoog basal salts mixture (Duchefa Biochemie) in 0.8% (w/v) agar at pH 5.8, and stratified in darkness at 4 °C for 2 days before transfer to Panasonic MLR-352 plant growth chambers. Cultivation occurred under cycles of 12 h light/12 h darkness at 19 °C, 90 µmol m$^{-2}$ s$^{-1}$ of white light, with experiments starting at a seedling age of 11 days. *SIG5* and *psbD* BLRP transcript abundance was measured after 3 and 5 h cold treatment, respectively, because there is a time delay between accumulation of *SIG5* transcripts and downstream *psbD* BLRP[24,26]. For gating experiments in the *hy5*, *hyh* and *hy5 hyh* backgrounds, both *SIG5* and *psbD* BLRP abundance was measured at the same time point (after 3 h of cold treatment). For single time point measurements of the response of transcripts to cold (Fig. 1), plants were transferred to the cold treatment 1 h after dawn. For circadian experiments (Fig. 2a,b and Extended Data Fig. 3a,b), seedlings were grown under light/dark cycles for 11 days and exposed to continuous light for 24 h before the start of the experiment. For measurements of chlorophyll fluorescence and protein abundance, seeds were sown directly onto compost and grown under cycles of 16 h light/8 h darkness at 20 °C, 90 µmol m$^{-2}$ s$^{-1}$ of white light for 14 days, before transfer to 4 °C in 12 h light/12 h darkness, 90 µmol m$^{-2}$ s$^{-1}$. For this experiment,

16 h days were used to increase the similarity of the experimental design to a previous study on HY5 and low-temperature responses[30]. The light spectrum was similar when the chambers were set to control and cold temperature conditions (Extended Data Fig. 8a,b; Li-Cor LI-180 spectrometer). For experiments using mature plants, seedlings were grown on MS agar as before and transferred to compost at 11 days, where they were grown for a further 24 days in a controlled environment chamber (Conviron) in 12 h light/12 h darkness at 19 °C, 100 µmol m$^{-2}$ s$^{-1}$ of white light before being used in experiments (light spectrum in Extended Data Fig. 8c). Experiments used the transfer DNA insertion mutant *sig5*-3[24] in the Col-0 background, and *hy5KS50* (*hy5*)[84], *hyh*, *hy5KS50 hyh* (*hy5 hyh*)[85] in the Wassilewskija (Ws) background. For investigation of roles for ATHB17, we used T-DNA insertion lines SALK_095524 (*athb17*-1)[27] and SALK_134535 (*athb17*-2) with Col-0 as the wild-type control.

### RNA extraction and RT–qPCR

Tissue for RNA isolation was snap frozen in liquid nitrogen and stored at -80 °C until RNA extraction. Frozen tissue was ground to a powder for RNA extraction using a Qiagen TissueLyzer II ball mill. Total RNA was isolated using the Macherey-Nagel Nucleospin II RNA extraction kit, using the aerial portion of ten *Arabidopsis* seedlings in each extraction[24,26]. Total RNA yield was always greater than 200 ng µl$^{-1}$, and any samples with $A_{260}/A_{280}$ below 2.0 were discarded (Thermo Fisher NanoDrop One). Complementary DNA was synthesized using the ABI High Capacity cDNA Reverse Transcription Kit (Thermo Fisher Scientific), using MultiScribe reverse transcriptase and random primers and with 1 µg of total RNA in each reaction. cDNA was analysed using Brilliant III Ultra-Fast SYBR Green QPCR master mix (Agilent Technologies) or qPCRBIO SyGreen (PCR Biosystems) and appropriate primer sets (Supplementary Data 7), normalized to *ACTIN2* using the delta-delta Ct (ddCT) method[24,26]. Analysis used a Bio-Rad CFX96 Touch Real Time PCR System (running Bio-Rad CFX v.3.1 software). We confirmed key results (response of *SIG5* to cold, variation in response to cold according to the time of day, alteration of the response to cold in *hy5 hyh* double mutant, equivalent response to cold in Col-0 and Ws backgrounds) using two further reference genes (*UBIQUITIN10* (*UBQ10*) and *EF1ALPHA* (*EF-1A*)[86]). Primers for analysis of *SIG5* and *psbD* BLRP transcript levels are from Noordally et al.[24], whereas those for *HY5* and *HYH* are from Hayes et al.[87] (Supplementary Data 7). We do not think our cold treatments caused a systematic change in chloroplast transcript levels—as might happen if the number or viability of chloroplasts was altered by cold—because there was not a systematic change in chloroplast transcript abundance detected by RNA sequencing (RNA-seq) analysis (Supplementary Data 3), as also reported elsewhere[10]. Transcript time-series data were analysed using the meta2d tool within MetaCycle[34], running in R v.4.1.1 (ref. 88) to identify rhythmic transcripts and properties of those rhythms.

### Data acquisition and analysis for RNA-seq

Seedlings were cultivated as for RT–qPCR analysis. RNA samples were collected from Col-0 and *sig5*-3 plants at two different time points. After 24 h under continuous light, one set of seedlings was exposed to 3 h of cold (4 °C), commencing at ZT45, and the other set was exposed to 5 h of cold, commencing at ZT29. These corresponded to times when *SIG5* (ZT45) and *psbD* BLRP (ZT29) transcripts accumulate strongly in response to chilling under free-running conditions (Fig. 2a,b). Total RNA was extracted from three replicates of ten seedlings each, using Macherey-Nagel Nucleospin II RNA extraction kits, and combined to represent one sample for sequencing. RNA concentrations were determined using a Nanodrop spectrophotometer (Thermo Fisher Scientific) and RNA integrity assessed using a Bioanalyzer (Agilent Technologies). Three independent biological replicates of these combined preparations were analysed by RNA-seq. The RNA-seq libraries were prepared from total RNA using an Illumina TruSeq Stranded mRNA kit, according to the manufacturer's instructions. Sequencing was performed using an Illumina NextSeq 500 using NSQ 500

Hi-Output kit v.2 (150 cycles). The quality of the sequencing was confirmed with FASTQC v.0.11.3 (ref. [89]). We trimmed any remaining adaptors using Trimmomatic v.0.33 (ref. [90]), with the flags PE -phred33 ILLUMINACLIP:TruSeq2-PE.fa:2:30:10 LEADING:20 TRAILING:20 SLIDINGWINDOW:10:20 MINLEN:50. The gene models for *Arabidopsis thaliana* (TAIR10_cds_20110103_representative_gene_model_updated) were downloaded from TAIR (https://www.arabidopsis.org, accessed 25 November 2019), and the counts were quantified with Kallisto v.0.44.0 (ref. [91]). The Kallisto gene counts were uploaded to Degust[92] for analysis. After identification of differentially expressed genes, the lists were filtered to include only transcripts with log(fold change) > 2 and $P \le 0.01$ (Voom/Limma method[46]). Analysis of RNA-seq data used R v.3.6.1. Gene names, descriptions and GO-term enrichment analysis were performed using the Thalemine tool of the Bio-Analytic Resource for Plant Biology (http://bar.utoronto.ca). We determined whether statistically significant overlaps existed between sets of transcripts by using a hypergeometric test, which considers whether the overlap between two sets of genes is significantly different from the size of an overlap arising from two randomly drawn sets of genes. This involves calculation of the representation factor, which is the actual number of genes in the intersection, divided by the expected number of genes in the intersection; thus a value >1 indicates a greater number of genes than expected, and a value <1 indicates fewer genes than expected. The probability of this intersection occurring was calculated using a normal approximation of the exact hypergeometric probability[93,94].

## Measurement of PSII photosynthetic efficiency and freezing survival

For measurement of PSII photosynthetic efficiency in seedlings, chlorophyll fluorescence parameters of seedlings were measured using an IMAGING-PAM M-series MAXI chlorophyll fluorescence imaging system (Walz; with Walz ImagingWin software v.2.47). Measurements were taken from 14-day-old plants grown at 20 °C. Plants were then transferred to 4 °C and measurements taken after 3, 24 and 48 h, and 7 and 10 days of cold treatment. For measurements after freezing, plants that had been acclimated at 4 °C for 10 days were transferred to −8 °C for 6 h in darkness, after which plants were placed at 4 °C to thaw. Measurements were taken approximately 18 h after freezing. For all experiments, plants were dark adapted for 20 min before fluorescence measurement. Measurements were initiated by exposing dark-adapted plants to measuring light pulses (frequency 1 Hz, intensity 3) and then applying a saturating pulse. Chlorophyll fluorescence parameters of mature rosette leaves were measured using a MINI-PAM (Walz). Leaves were dark adapted for 20 min before applying a saturating light pulse. $F_v/F_m$ was calculated as $(F_m − F_0)/F_m$. For assessment of freezing survival 14-day-old plants were acclimated at 4 °C for 10 days (acclimation started at dawn), then placed at −8 °C for 6 h (Percival Intellus LT-36VL chamber, CLF Plant Climatics). Survival was assessed after a 7-day recovery period at 20 °C, with recovery occurring under growth conditions described previously.

## Protein extraction and immunodetection

For protein abundance analysis, plants were grown as described for measurement of PSII photosynthetic efficiency and collected from the same experimental material. Samples were taken after 14 days of growth and after 10 days at 4 °C followed by 6 h at −8 °C with 18 h recovery. Protein extraction was conducted as described previously[95]. Briefly, powdered tissue was incubated in protein extraction buffer (50 mM Tris−HCl pH 7.5, 150 mM NaCl, 5 mM dithiothreitol, protease inhibitor cocktail (Sigma) 1:100, phosphatase inhibitor (Sigma) 1:200, 1 mM phenylmethylsulfonylfluoride, 0.5% IPEGAL CA-630 (Sigma), 1 mM EDTA, 1 mM $Na_2MoO_4 \times 2H_2O$, 1 mM NaF, 1.5 mM activated $Na_3VO_4$) at 4 °C for 1 h, and supernatant isolated by centrifugation. Total protein concentration was normalized to 0.5 mg ml$^{-1}$ using a Bradford assay (Sigma reagent B6916 for protein range 0−1.4 mg ml$^{-1}$; calibrated across protein concentration range 0−1.2 mg ml$^{-1}$ using a BSA standard), and confirmed using Coomassie

blue staining after SDS−PAGE (Extended Data Fig. 5c). Protein abundance was analysed using an automated semiquantitative capillary immunoassay (ProteinSimple Wes; running ProteinSimple Compass software v.4.1.0)[96–98] with a Wes-Rabbit (2–40 kDa) Master kit according to the manufacturer's instructions. Samples were loaded on Wes cartridges, running three technical replicates per sample, along with a polyclonal rabbit anti-PSII D2 antibody diluted 1:5,000 (AS06146), a rabbit anti-PSAC antibody diluted 1:1,000 (AS10939), and a rabbit anti-RbcL antibody diluted 1:5,000 (AS03037, Agrisera). Experiments were repeated twice.

## Electrolyte leakage assays

Cellular damage after freezing was quantified by measuring the leakage of electrolytes from tissue that had been frozen, as a proxy for freezing damage, using a method similar to that used by Hemsley et al.[99]. Mature plants were grown as described previously for 5 weeks, after which freezing tolerance was analysed. For analysis of cold-acclimated plants, 5-week-old plants were transferred to a Sanyo MLR-352 under cycles of 12 h light/12 h darkness at 4 °C, 90 μmol m$^{-2}$ s$^{-1}$ of white light for 2 weeks, before the freezing treatments were applied. Five biological replicate samples per genotype per freezing temperature tested were prepared. Rosette leaves for analysis were removed and washed with deionized water to remove ionic material from the leaf surface. Three 8-mm leaf discs were obtained from each plant using a cork borer and placed in a glass test tube held on ice. After preparation of all tubes, a set of control tubes was held on ice, while the freezing treatment tubes were transferred to a sub-zero water bath with an immersion dip cooler. After 2 h at −2 °C, deionized water ice chips were added to tubes to initiate ice nucleation. The temperature of the water bath was reduced progressively to each test temperature, with sets of tubes moved to ice 30 min after each test temperature was reached. Samples were thawed gradually on ice overnight, and 5 ml of deionized water was added to each tube. Tubes were shaken for 3 h at room temperature. The electrical conductivity of the water in the tube was measured using a conductivity meter (Mettler Toledo Seven2Go). The tubes with leaves were then frozen to −80 °C for 1 h, to release the solutes that remained within the tissue. These tubes were shaken for 3 h at room temperature, and the electrical conductivity measured again. Electrolyte leakage was calculated as the proportion of electrolytes released at each freezing temperature, relative to the total electrolytes present in the samples. Each experiment was repeated independently three times.

## Transient luciferase expression by particle bombardment

A *SIGS::LUCIFERASE* reporter construct (pGREENII0229 SIG5::LUCIFERASE[24]) was expressed transiently in *Arabidopsis* seedlings using methods similar to those used previously[35,36]. Briefly, 5 μg of pGREENII0229 SIG5::LUCIFERASE or pB7WG2.0-GFP (green fluorescent protein (GFP) positive control for transformation[100]) was combined with 25 μl of 1 nm gold particle suspension (Bio-Rad), 25 μl of 2.5 M $CaCl_2$ and 10 μl of 0.1 M spermidine (Sigma), and incubated on ice for 30 min. DNA-coated gold particles were washed with 100% ethanol, resuspended in 100% ethanol and stored at −20 °C before particle bombardment. A Bio-Rad PDS-1000/He particle delivery system was used for bombardment of *Arabidopsis* plants, with a 1,350 p.s.i. rupture disk. Eleven-day-old seedlings, cultivated as for the RT−qPCR experiments, were positioned at the closest distance position to the gun muzzle ('floor 2') for bombardment. After bombardment, 5 mM luciferin (D-luciferin potassium salt; Melford Laboratories) was applied to plants using a small spray bottle 24 h before imaging commenced. A Photek HRPCS-intensified CCD photon counting system (Photek Ltd, with Photek Image32 software) was used to image luciferase bioluminescence, using 10-min integrations, with the camera set to photon counting mode. The first 90 s of data was discarded to eliminate interfering chlorophyll autofluorescence (delayed fluorescence). Plants were under constant light conditions between the two time points measured. The GFP transformation control reporter was imaged using a LeicaM205FA fluorescence microscope around 48 h after bombardment.

## Reporting summary

Further information on research design is available in the Nature Portfolio Reporting Summary linked to this article.

## Data availability

The RNA-seq data for this study are available in the European Nucleotide Archive (ENA; https://www.ebi.ac.uk/ena) with the project ID PRJEB45855. All other data supporting the findings of this study are included in the main figures, extended data figures and supplementary information.

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

## Acknowledgements

This research was funded by Biotechnology & Biological Sciences Research Council (UK) (BB/I005811/2, BB/J014400/1, BB/T003030/1, Institute Strategic Programme GEN BB/P013511/1 to A.N.D.; studentship 1518540 awarded to P.E.P.), Norwich Research Park Doctoral Training Partnership (BB/T008717/1, to R.D. and A.B.), The Leverhulme Trust (RPG-2018-216, to A.N.D.), the Bristol Centre for Agricultural Innovation (to A.N.D.), the Wolfson Foundation (to A.N.D.), NAGASE Science Technology Foundation (Japan), the Ministry of Education, Culture, Sports, Science and Technology (Japan) (Grants-in-Aid 17K07438, to K.T. and S.I.) and Tokyo Institute of Technology World Research Hub Initiative Program of Institute of Innovative Research (to K.T.). D.L.C.-R. is grateful to the Consejo Nacional de Ciencia y Tecnología (Mexico) for granting a PhD scholarship. We thank the University of Bristol Genomics Facility and Z. Song for experimental support; G. Jenkins for seed donation; S. Samwald, E. Tee, J. Sallmen and N. Holmes for help with protein analysis; C. Faulkner, T. Oyama and T. Muranaka for advice about transient expression; and M. Knight, Y. Yoshitake and M. Shimojima for technical advice. Figure 5 created with BioRender.com.

## Author contributions

D.L.C.-R. and A.N.D. conceived the study. D.L.C.-R., P.E.P., T.T., S.I., P.P., T.S.d.F., L.L.d.B.D., R.D., A.B., T.B.-G., B.F.M., K.A.F., H.K. and A.N.D. performed experimentation, data analysis and data interpretation. T.B. supported bioinformatic analysis. D.L.C.-R., P.E.P., P.P., L.L.d.B.D., K.T., S.I., K.A.F., H.K. and A.N.D. wrote the paper.

## Competing interests

The authors declare no competing interests.

## Additional information

**Extended data** is available for this paper at https://doi.org/10.1038/s41477-023-01377-1.

**Correspondence and requests for materials** should be addressed to Antony N. Dodd.

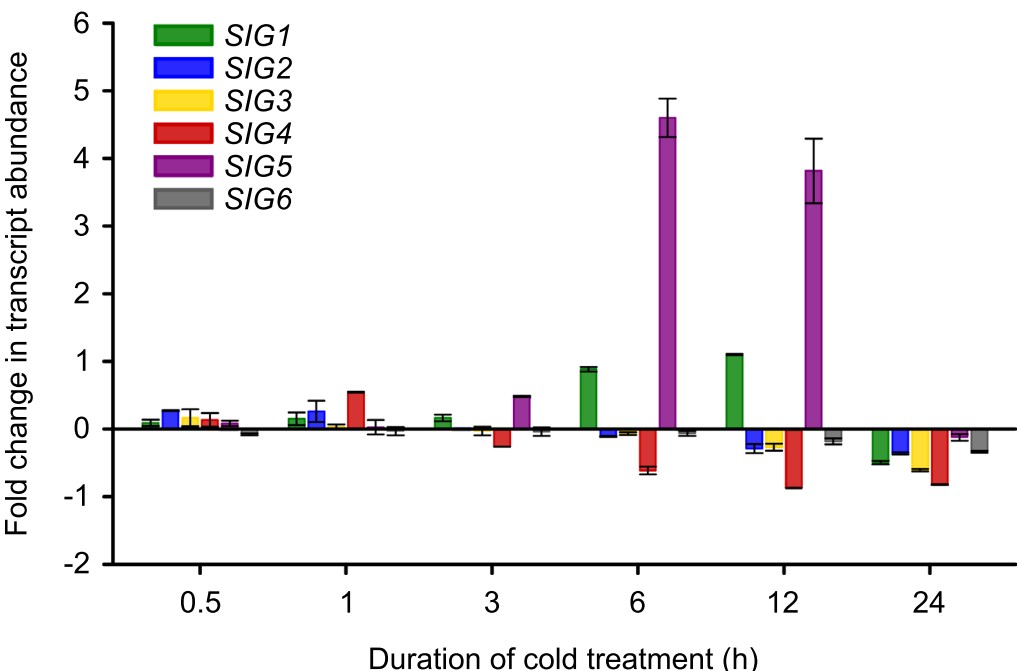

**Extended Data Fig. 1 | Transcriptome data identifies SIG5 transcript accumulation in response to cold.** Transcriptome data identifies *SIG5* transcript accumulation in response to cold. Data indicate the fold-change in abundance of all six Arabidopsis sigma factors (*SIG1-6*) during a prolonged cold treatment, measured using microarray analysis in[28]. Data extracted using the Arabidopsis eFP Browser (https://bar.utoronto.ca)[101]. Data are means from two biological replicates +/- s.d., as described in[28].

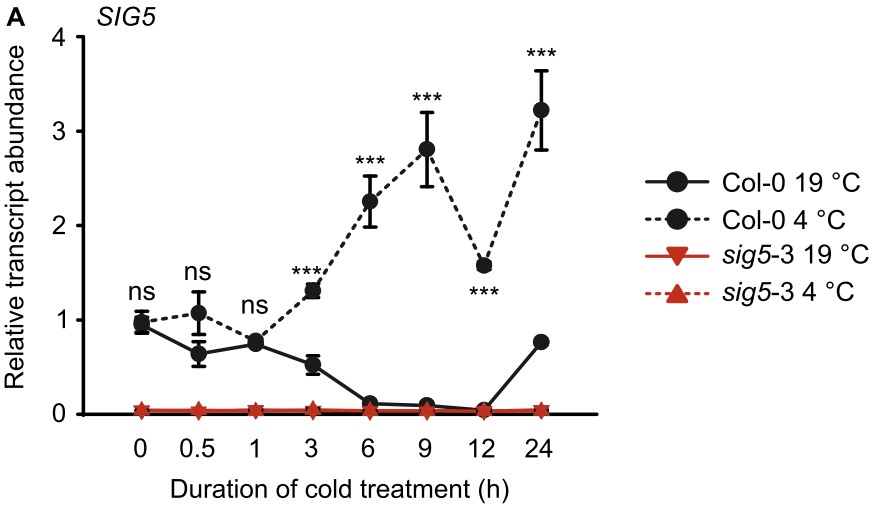

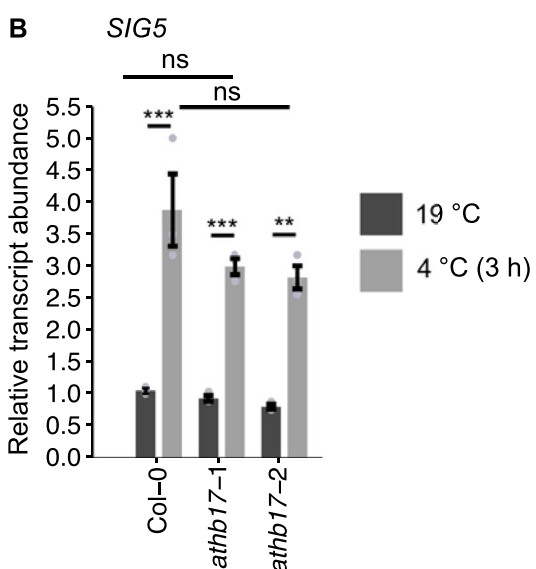

**Extended Data Fig. 2 | SIG5 transcript accumulation during cold treatments of up to 24 h.** SIG5 transcript accumulation during cold treatments of up to 24 h. (A) Relative abundance of *SIG5* transcripts in Col-0 wild type and *sig5*-3 plants after cold treatment (4 °C) for durations ranging from 30 mins to 24 hours. (B) Abundance of *SIG5* transcripts in wild type (Col-0) and two *athb17* mutants after 3 h at 19 °C or 4 °C. Plants were under constant light conditions and given a 3 h cold treatment 1 h after subjective dawn. Experiments used 11-day old seedlings. Data are mean +/- s.e.m. (n = 3 independent biological replicates). Statistical comparisons are of (A) transcript abundance between Col-0 19 °C and

Col-0 4 °C and (B) transcript abundance between control temperature and cold-treated plants, and between genotypes. Data analysed by (A) one-way ANOVA and (B) two-way ANOVA, both followed by one-sided post-hoc Tukey test, where *** = p < 0.001; ** = p < 0.01, * = p < 0.05 and n.s = not significant. p-values (A) 19 °C *vs* 4 °C for Col-0 at 0 h (p = 1.0000), 0.5 h (p = 0.9277), 1 h (p = 1.0000), 3 h (p = 0.1867), 6 h (p < 0.0001), 9 h (p < 0.0001), 12 h (p = 0.0001), 24 h (p < 0.0001); (B) 19 °C *vs* 4 °C for Col-0 (p = 0.0000425), *athb17*-1 (p = 0.0008523); *athb17*-2 (p = 0.0010056); Col-0 *vs athb17*-1 at 4 °C (p = 0.1958629); Col-0 *vs athb17*-2 at 4 °C (p = 0.0918899).

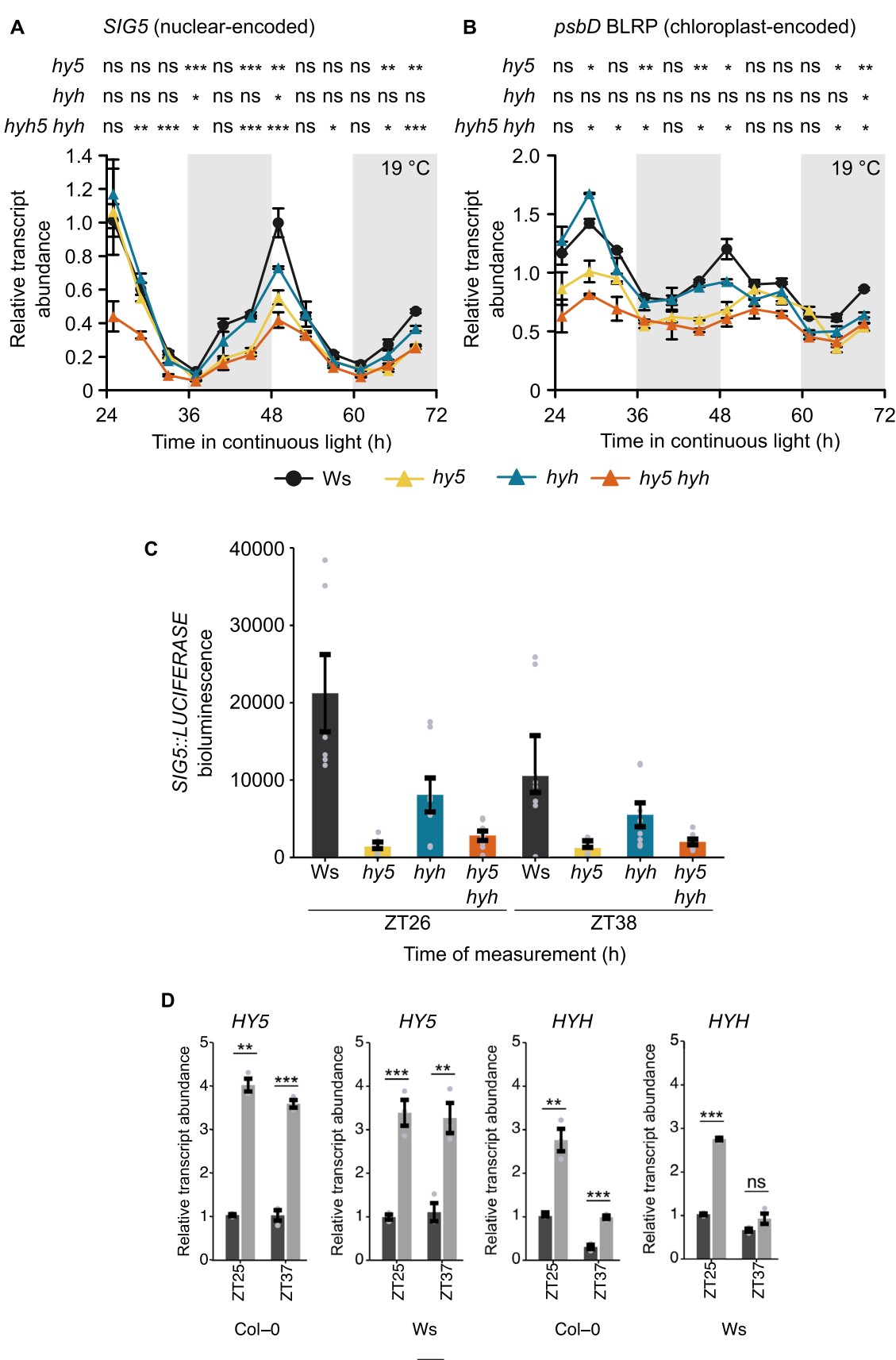

**Extended Data Fig. 3 | See next page for caption.**

**Extended Data Fig. 3 | Accumulation of *SIG5* and *psbD* BLRP transcripts in the wild type (Ws), *hy5*, *hyh* and *hy5 hyh* double mutant.** Accumulation of (A) *SIG5* and (B) *psbD* BLRP transcripts in the wild type (Ws), *hy5*, *hyh* and *hy5 hyh* double mutant. Experiments used 11-day old seedlings. All data represent means +/- s.e.m. of three independent biological replicates. Statistical comparison of each mutant with the wild type, at each timepoint, is provided above time-series plots, where *** = p < 0.001; ** = p < 0.01, * = p < 0.05 and n.s = not significant (one-way ANOVA followed by one-sided post-hoc Tukey test; 3 biological replicates; p values for panels A and B in Supplementary Data 1). (C) Quantification of luciferase bioluminescence after transient expression of *SIG5::LUCIFERASE*

in wild type, *hy5*, *hyh* and *hy5 hyh mutants*, at two timepoints, using particle bombardment (n = 8 replicates per condition and genotype, except for Ws at ZT26 where n = 6; mean +/- s.e.m.). (D) Response of *HY5* and *HYH* transcripts to a treatment of 3 h at 4 °C, given at two different timepoints, under free running conditions. Transcript levels normalized to *ACT2* and confirmed with two other reference transcripts (not shown), 3 biological replicates show as mean +/- s.e.m; *** = p < 0.001; ** = p < 0.01, * = p < 0.05 and n.s = not significant in unpaired two-sided t-tests. p values for (D) are Col-0 *HY5* (ZT25 0.002; ZT37 0.000), Ws *HY5* (ZT25 0.001; ZT37 0.006), Col-0 *HYH* (ZT25 0.003; ZT37 0.000), Ws *HYH* (ZT25 0.000; ZT37 0.098).

**A**

Representation factor = 12.2
Significant overlap, p=5.207e-300

ZT29

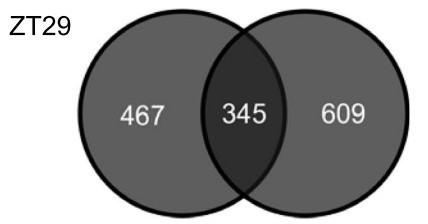

Zhao et al. 2016          This study

Representation factor = 13.9
Significant overlap, p=1.596e-95

ZT45

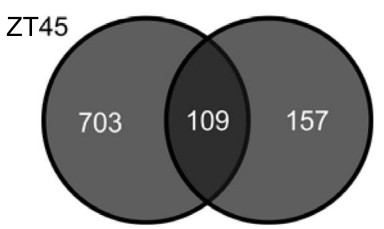

Zhao et al. 2016          This study

**B**

Representation factor = 0.6
Significant 'underlap', p=9.13e-4

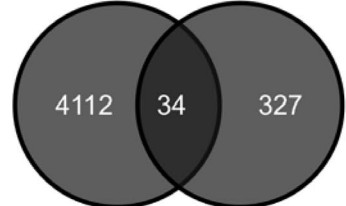

Circadian
Covington et. al. (2008)          Cold-responsive
Edwards et. al. (2006)          in Col-0 only

Representation factor = 1.1  (p=0.289)

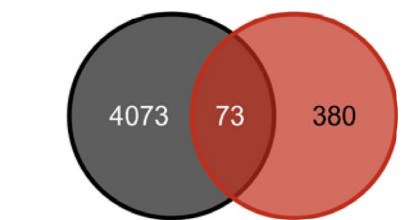

Circadian
Covington et. al. (2008)          Cold-responsive
Edwards et. al. (2006)          in sig5-3 only

**C**

Representation factor = 1.1 (p=0.245)

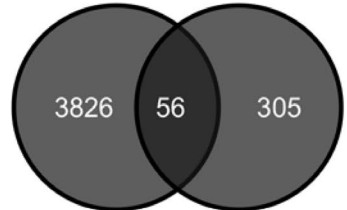

HY5 putative targets          Cold-responsive
Lee et. al (2007)          in Col-0 only

Representation factor = 1.4
Significant overlap, p=0.002

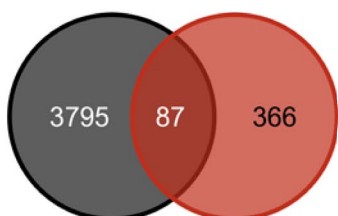

HY5 putative targets          Cold-responsive
Lee et. al. (2007)          in sig5-3 only

**D**

Representation factor = 1.1 (p=0.467)

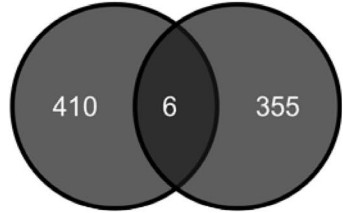

HY5 cold-regulated          Cold-responsive
Catala et al. (2011)          in Col-0 only

Representation factor = 0.7 (p=0.315)

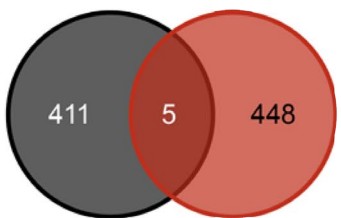

HY5 cold-regulated          Cold-responsive
Catala et al. (2011)          in sig5-3 only

**Extended Data Fig. 4 | RNA sequencing analysis of transcripts in Col-0 wild type and *sig5*-3 that respond to a cold treatment at two different times of day.** RNA sequencing analysis of transcripts in Col-0 wild type and *sig5*-3 that respond to a cold treatment at two different times of day. (A) Overlap between cold-responsive transcripts in Col-0 from this study and cold-responsive transcriptome from[48]. (B) Overlap between cold-responsive transcripts in Col-0 or *sig5*-3 from this study and circadian-regulated transcripts from[50] and[49].

(C) Overlap between cold-responsive transcripts in Col-0 or *sig5*-3 from this study and putative HY5 targets[37]. (D) Overlap between cold-responsive transcripts in Col-0 or *sig5*-3 from this study and transcripts regulated in response to cold by HY5[30]. Statistical significance and representation factors were calculated using a hypergeometric test (one-sided; see methods). Experiments used 11-day old seedlings.

**A** PSII D2

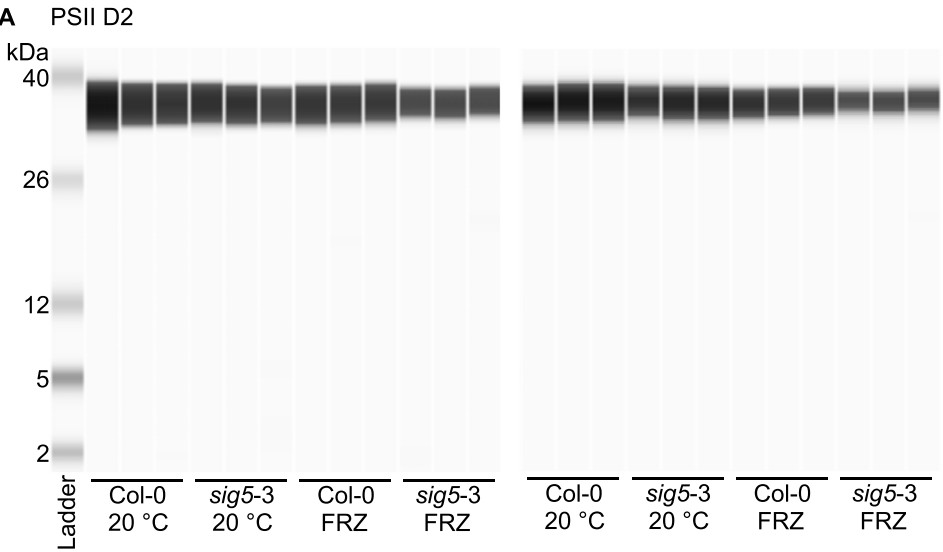

**B** PSAC

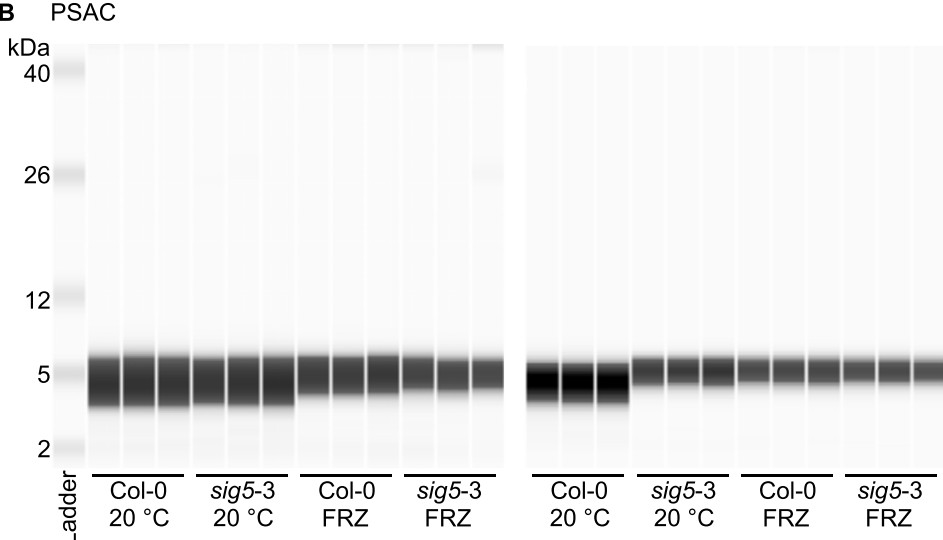

**C**

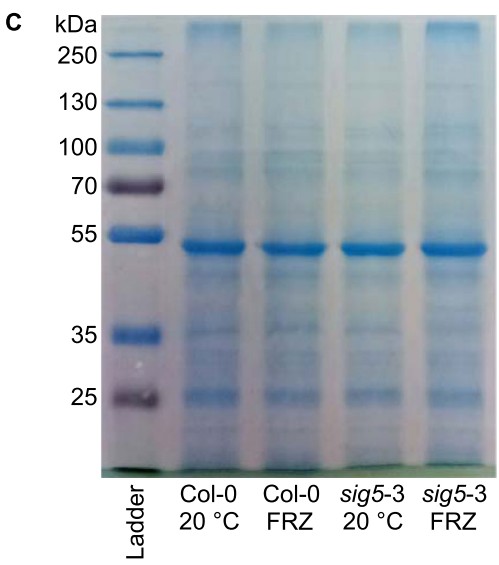

**Extended Data Fig. 5 | See next page for caption.**

**Extended Data Fig. 5 | Altered photosystem protein abundance in *sig5*-3 mutant.** Altered photosystem protein abundance in *sig5*-3 mutant. (A, B) Automated semi-quantitative immunoassay comparing (A) PSII D2 and (B) PSAC protein abundance between wild type (Col-0) and *sig5*-3 under control temperature conditions (20 °C) and after freezing at -8 °C for 6 h (FRZ).

Analysis shows two independent experiments, each containing triplicate immunodetection analyses. (C) Coomassie blue-stained SDS-PAGE separation of 0.5 mg/mL total protein from leaf protein extracts, run as a single example to demonstrate consistent protein input into the automated semi-quantitative immunoassay when samples were prepared identically for immunodetection.

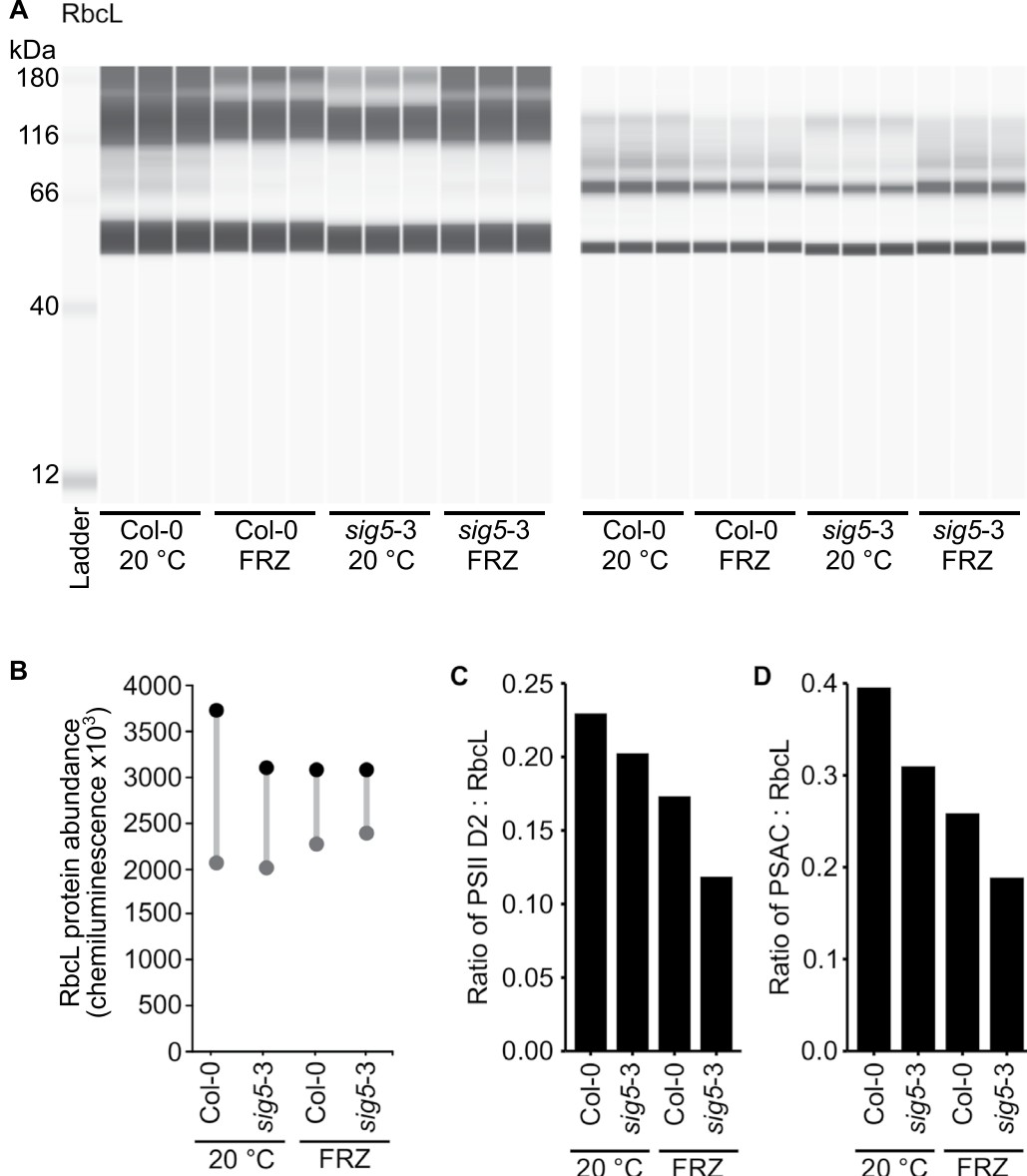

**Extended Data Fig. 6 | RbcL protein abundance is unaltered by *sig5*-3 mutant and freezing.** RbcL protein abundance is unaltered by *sig5*-3 mutant and freezing. (A) Automated semi-quantitative immunoassay comparing RbcL protein abundance between wild type (Col-0) and *sig5*-3 under control temperature conditions (20 °C) and after freezing at -8 °C for 6 h (FRZ). Analysis shows two independent experiments, each containing triplicate immunodetection analyses. (B) Comparison of RbcL protein abundance from (A, B). Circles on plots indicate result from each independent experiment (two independent repeats, each with three technical replicates). (C, D) Ratio of (C) mean PSII D2 abundance and (D) mean PSAC abundance to mean RbcL abundance, calculated from the protein abundance data in Fig. 4D, F and Extended Data Fig. 6B.

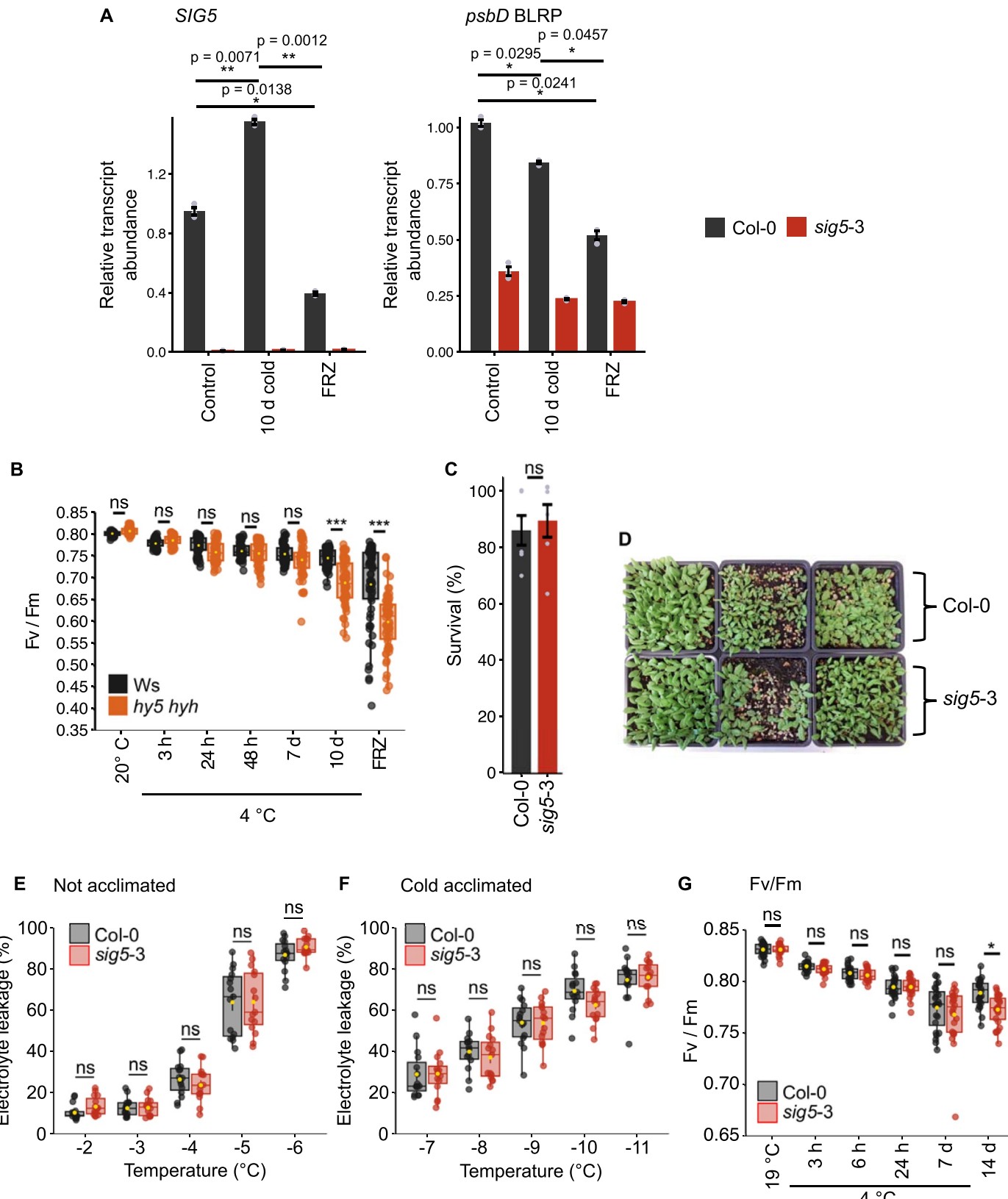

**Extended Data Fig. 7 | See next page for caption.**

**Extended Data Fig. 7 | Responses to low and freezing temperature conditions in the *sig5*-3 mutant.** Responses to low and freezing temperature conditions in the *sig5*-3 mutant. (A) Abundance of *SIG5* and *psbD* BLRP transcripts in the wild type (Col-0) and *sig5*-3 mutant after 10 days at 4 °C and after exposure to freezing (FRZ) conditions. Conditions were comparable to those shown in Fig. 4A and C (n = 3 independent repeats, mean +/- s.e.m.). (B) Photosynthetic efficiency of PSII (Fv/Fm) of 14-day old wild type (Ws) and *hy5 hyh* plants exposed to cold and freezing (FRZ) treatments (n = 60). (C) Survival of Col-0 wild type and *sig5*-3 plants grown for 14 days at 20 °C then acclimated at 4 °C for 10 days. Plants were subjected to -8 °C for 6 h and then allowed to recover at 20 °C for 7 days (n = 6; data analysed by Student's t-test; two-sided; not significant; mean +/- s.e.m.). (D) Representative appearance of Col-0 and *sig5*-3 plants after 7 days of recovery at 20 °C after the freezing treatment, showing variation across three replicate pots per genotype. (E, F) Electrolyte leakage after freezing of leaf discs from plants of wild type (Col-0) and *sig5*-3. Plants were tested after 5 weeks of growth (E, not acclimated) or 5 weeks of growth plus 2 weeks of acclimation at 4 °C (F, cold acclimated). n = 15. (G) Photosynthetic efficiency of PSII (Fv/Fm) of 5-week-old Col-0 and *sig5*-3 plants after treatment at 19 °C or 4 °C for 3, 6, 24 hours, 7 and 14 days (n = 24, mean +/- s.e.m; p-values of Col-0 *vs sig5*-3 are p = 1.0000 (19 °C), p = 0.9999 (3 h), p = 0.9999 (6 h), p = 1.0000 (24 h), p = 0.7955 (7 days), p = 0.0011 (14 days)). (A, B, E-G) Data analysed by two-way ANOVA followed by post-hoc one-sided Tukey test, with (E, F) arcsine correction before analysis. *** = p < 0.001; ** = p < 0.01, * = p < 0.05 and n.s = not significant. On boxplots (B, E-G), box indicates interquartile zone with median line at the centre, whiskers indicate interquartile range, and yellow dot indicates the mean.

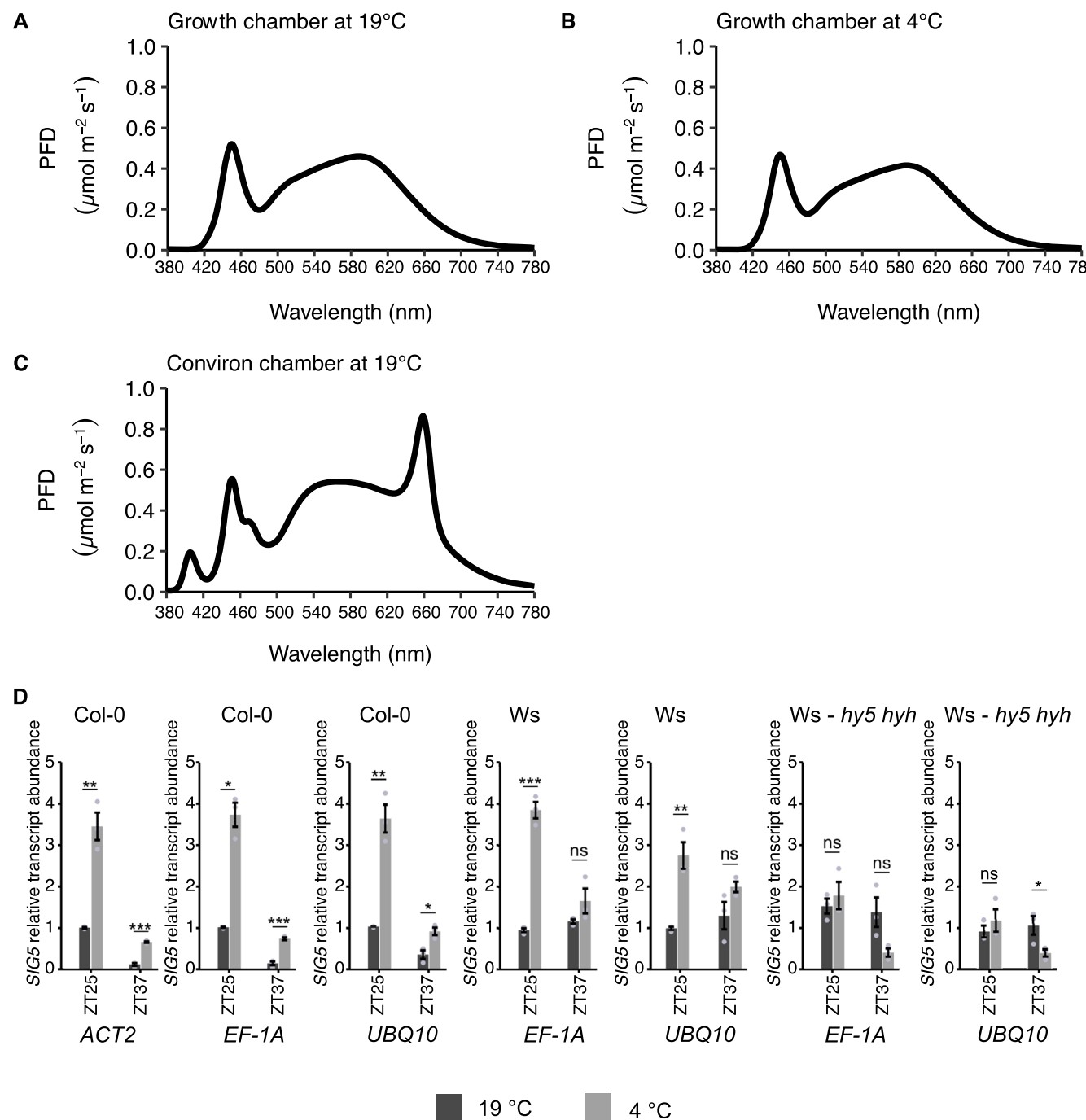

**Extended Data Fig. 8 | Light spectra and RT-qPCR controls.** Light spectra and RT-qPCR controls. (A, B) Comparison of spectra in Panasonic MLR-352 growth chambers set to (A) control and (B) cold temperature conditions. (C) Spectrum of Conviron growth chambers used to propagate mature plants. (D) Comparison of response of *SIG5* to cold, at two different timepoints, in two background lines, and in the *hy5 hyh* double mutant. The transcript dynamics are broadly similar when each of *ACT2*, *EF-1A* and *UBQ10* was used as a reference gene. *** = p < 0.001;

** = p < 0.01, * = p < 0.05 and n.s = not significant in unpaired two-sided t-tests, n = 3 biological replicates shown as mean +/- s.e.m. p values: Col-0 *ACT2* (ZT25 0.002; ZT37 0.000), Col-0 *EF-1A* (ZT25 0.011; ZT37 0.000), Col-0 *UBQ10* (ZT25 0.002; ZT37 0.014), Ws *EF-1A* (ZT25 0.000; ZT37 0.239), Ws *UBQ10* (ZT25 0.006; ZT37 0.239), *hy5 hyh EF-1A* (ZT25 0.534; ZT37 0.57), *hy5 hyh UBQ10* (ZT25 0.432; ZT37 0.048).

# Reporting Summary

## Statistics

For all statistical analyses, confirm that the following items are present in the figure legend, table legend, main text, or Methods section.

| n/a | Confirmed | |
|---|---|---|
| ☐ | ☒ | The exact sample size ($n$) for each experimental group/condition, given as a discrete number and unit of measurement |
| ☐ | ☒ | A statement on whether measurements were taken from distinct samples or whether the same sample was measured repeatedly |
| ☐ | ☒ | The statistical test(s) used AND whether they are one- or two-sided *Only common tests should be described solely by name; describe more complex techniques in the Methods section.* |
| ☒ | ☐ | A description of all covariates tested |
| ☒ | ☐ | A description of any assumptions or corrections, such as tests of normality and adjustment for multiple comparisons |
| ☐ | ☒ | A full description of the statistical parameters including central tendency (e.g. means) or other basic estimates (e.g. regression coefficient) AND variation (e.g. standard deviation) or associated estimates of uncertainty (e.g. confidence intervals) |
| ☐ | ☒ | For null hypothesis testing, the test statistic (e.g. $F$, $t$, $r$) with confidence intervals, effect sizes, degrees of freedom and $P$ value noted *Give P values as exact values whenever suitable.* |
| ☒ | ☐ | For Bayesian analysis, information on the choice of priors and Markov chain Monte Carlo settings |
| ☒ | ☐ | For hierarchical and complex designs, identification of the appropriate level for tests and full reporting of outcomes |
| ☒ | ☐ | Estimates of effect sizes (e.g. Cohen's $d$, Pearson's $r$), indicating how they were calculated |

*Our web collection on statistics for biologists contains articles on many of the points above.*

## Software and code

Policy information about availability of computer code

| Data collection | RT-qPCR data were acquired with Bio-Rax CFX v3.1<br>Chlorophyll fluorescence imaging data were acquired and analyzed with Walz ImagingWin v2.47<br>Protein abundance data were acquired and analyzed with ProteinSimple Compass v4.1.0<br>Bioluminescence imaging data were acquired and analyzed with Photek Image32 |
|---|---|
| Data analysis | RNAseq analysis used Fastqc 0.11.3, Trimmomatic 0.33, Kallisto 0.44.0, Degust (DOI: 10.5281/zenodo.3258932; degust.erc.monash.edu). Circadian time-series analysis used the meta2d command within the MetaCycle v1.2.0 package, running in R v4.1.1. Data analysis and graphs prepared using R v4.1.1 (using ggplot2 v3.4.0, ggpubr v0.4.0 and cowplot v1.1.1); RNA sequencing data analyzed using R v3.6.1. |

For manuscripts utilizing custom algorithms or software that are central to the research but not yet described in published literature, software must be made available to editors and reviewers. We strongly encourage code deposition in a community repository (e.g. GitHub). See the Nature Portfolio guidelines for submitting code & software for further information.

## Data

Policy information about availability of data

All manuscripts must include a data availability statement. This statement should provide the following information, where applicable:

- Accession codes, unique identifiers, or web links for publicly available datasets
- A description of any restrictions on data availability
- For clinical datasets or third party data, please ensure that the statement adheres to our policy

The RNASeq data for this study are available in the European Nucleotide Archive (ENA; https://www.ebi.ac.uk/ena) with the project ID PRJEB45855.

# Field-specific reporting

Please select the one below that is the best fit for your research. If you are not sure, read the appropriate sections before making your selection.

☒ Life sciences          ☐ Behavioural & social sciences          ☐ Ecological, evolutionary & environmental sciences

For a reference copy of the document with all sections, see nature.com/documents/nr-reporting-summary-flat.pdf

# Life sciences study design

All studies must disclose on these points even when the disclosure is negative.

| | |
|---|---|
| Sample size | No sample-size calculation was performed. Sample sizes were based on sample sizes used previously to investigate this signalling pathway (Noordally et al. Science 2013 (https://doi.org/10.1126/science.1230397), Belbin et al. New Phytologist 2016 (https://doi.org/10.1111/nph.14176). |
| Data exclusions | No data were excluded from analyses. |
| Replication | At least three completely independent experiments were conducted for each dataset, with the exception of protein abundance experiments where two completely independent experiments were conducted. All replication attempts across all experiments were successful (no data are included that did not replicate successfully). |
| Randomization | Plant material was positionally-randomized within growth chambers. |
| Blinding | Blinding was not used for temperature treatments (it is not possible to blind for time of day or temperature treatment, because this is obvious to the experimentalist). Experimentalists were blind to genotype during RNA isolation, RT-qPCR, protein isolation and immunodetection, and during physiological assays (chlorophyll fluorescence analysis, freezing tolerance analysis). |

# Reporting for specific materials, systems and methods

We require information from authors about some types of materials, experimental systems and methods used in many studies. Here, indicate whether each material, system or method listed is relevant to your study. If you are not sure if a list item applies to your research, read the appropriate section before selecting a response.

## Materials & experimental systems

| n/a | Involved in the study |
|---|---|
| ☐ | ☒ Antibodies |
| ☒ | ☐ Eukaryotic cell lines |
| ☒ | ☐ Palaeontology and archaeology |
| ☒ | ☐ Animals and other organisms |
| ☒ | ☐ Human research participants |
| ☒ | ☐ Clinical data |
| ☒ | ☐ Dual use research of concern |

## Methods

| n/a | Involved in the study |
|---|---|
| ☒ | ☐ ChIP-seq |
| ☒ | ☐ Flow cytometry |
| ☒ | ☐ MRI-based neuroimaging |

## Antibodies

| | |
|---|---|
| Antibodies used | Antibodies used were for PSII D2 (Agrisera AS06146); PSI-C (Agrisera AS10939); RbcL (Agrisera AS03037). |
| Validation | All three antibodies are validated by Agrisera with this information provided on the supplier's website. Dilution series of antibodies were conducted before experimentation to ensure optimal specificity. PSII D2 (Agrisera AS06146), PSI-C (PSAC) (AS10939) and RbcL (AS03037) validated by manufacturer for use with Arabidopsis thaliana. |

