## [Peer Review File · Nature Plants]

Peer Review Information

Journal: Nature Plants

Manuscript Title: Low temperature and circadian signals are integrated by the sigma factor SIG5

Corresponding author name(s): Professor Antony Dodd

Reviewer Comments & Decisions:

Decision Letter, initial version:

Message 29th June 2022

:

Dear Professor Dodd,

Your Article, "Integration of low temperature and circadian signals by a plant sigma factor" has now been seen by three referees. You will see from their comments below that while they find your work of interest, some important points are raised. We are interested in the possibility of publishing your study in Nature Plants, but would like to consider your response to these concerns in the form of a revised manuscript before we make a final decision on publication.

As you can read below, referees #2 and #3 are overall positive and convinced that the work presents interesting data. Referee #1 is more concerned that the study is in a premature state and gives substantial advice for improvements. In our opinion, the main issues raised by the referees are:

1. Reviewer #1 criticizes the use of actin-2 as a control protein (affected by cold, different dynamics than plastid proteins) and suggest the use of additional controls. Experiments with SIG5 overexpression lines are suggested to support some of the conclusions.
2. Reviewer #2 suggests effector-reporter assays to test direct activity of HY5/HYH on the SIG5 promoter.
3. Reviewer #3 is concerned that the activity of PEP itself could have an effect on chloroplast gene expression independent of any sigma factor. This could be tested by run-on or ChIP-seq analyses. The indirect effects of SIG5 on nuclear gene expression could be tested in retrograde signaling mutants. Conclusions about effects of long-term cold and freezing would require more data on transcript accumulation under these conditions.

All referees also raise some minor points as well as suggestions for textual changes that should be addressed. As usual, please do not hesitate to get in touch if you would like to discuss these issues further.

We therefore invite you to revise your manuscript taking into account all reviewer and editor comments. Please highlight all changes in the manuscript text file, preferably in Microsoft Word format.

When revising your manuscript please:

* If you have not done so already please begin to revise your manuscript so that it

2conforms to our Article format instructions at <http://www.nature.com/nplants/info/final-submission> . Refer also to any guidelines provided in this letter.

* Pay close attention to our [href="https://www.nature.com/nature-portfolio/editorial-policies/image-integrity">Digital Image Integrity Guidelines. and to the following points. Please ensure:](https://www.nature.com/nature-portfolio/editorial-policies/image-integrity)

- that unprocessed scans are clearly labelled and match the gels and western blots presented in figures.
- that control panels for gels and western blots are appropriately described as loading on sample processing controls
- that all images in the paper are checked for duplication of panels and for splicing of gel lanes.
- that you retain unprocessed data and metadata files after publication, ideally archiving data in perpetuity. These may be requested during the peer review and production process or after publication if any issues arise.

Nature Plants strongly supports public availability of data and are therefore keen that the data used in your paper is placed in an appropriate public data repository. Alternatively, if this is not possible, you may present the data as Extended Data or Supplementary Information. If data can only be shared on request, please explain why in your Data Availability Statement, and also in the correspondence with your editor. Please note that for some data types, deposition in a public repository is mandatory.

[redacted]

We hope to receive your revised manuscript within two to three months (or more, this is not a strict deadline). If you think you will need more time, please let us know. We will be happy to consider your revision so long as nothing similar has been accepted for publication at Nature Plants or published elsewhere.

Sincerely,

[redacted]

Nature Plants is committed to improving transparency in authorship. As part of our

efforts in this direction, we are now requesting that all authors identified as 'corresponding author' on published papers create and link their Open Researcher and Contributor Identifier (ORCID) with their account on the Manuscript Tracking System (MTS), prior to acceptance. This applies to primary research papers only. ORCID helps the scientific community achieve unambiguous attribution of all scholarly contributions. You can create and link your ORCID from the home page of the MTS by clicking on 'Modify my Springer Nature account'. For more information please visit www.springernature.com/orcid.

Reviewers' Comments:

Reviewer #1:

Comments for the Author:

The manuscript „Integration of low temperature and circadian signals by plant sigma factor“ by Cano-Ramirez describes cold induction of SIG5 and shifted hy5 and hyh-independent, but hy5 hyh dependent cold-regulation of one of its putative target genes psbD and similar regulation of one non-target gene, psaC, in Arabidopsis thaliana. The manuscript is interesting, but in a premature state.

Concerns:

(1) The manuscript describes its results without careful consideration of the knowledge of our time. It hardly reflects the literature. For example, the statement "we know relatively little about how information concerning low temperature conditions is communicated to chloroplasts" (abstract) is simply wrong. We know a lot. One of the best studied examples are cold regulation of the expression of the chloroplast membrane protein Cor15A (external sensing) and cold regulation of photosynthesis (internal sensing). We also know that SIG5 has (like almost all genes) a complex promoter with binding sites for various transcriptional regulators (= more than HY5, HYH and ATHB17) and responds to various abiotic and biotic stress treatments (see e.g. eFP browser). The introduction is in the present form very, very short and lacks lot of information the reader requires to understand the experiments and the results.

(2) HY5 and HYH control photomorphogenesis. In many photomorphogenic and signaling processes, lack of HY5 can cause the phenotype on its own. Here, lack of HY5 and lack of HYH, both, had on their own no cold effect on SIG5. Cold-induction of SIG5 was only disturbed in hy5 hyh double mutants. Such combinatory effect is unusual and should have been discussed in more detail.

Photomorphogenesis has also a strong impact on chloroplast development and might feed-back on SIG5. Such an effect can cause un- (less) specific regulation, such as the described co-regulation of psaC and psbD and asks for more careful experiment design. In the discussion, the authors themselves present doubts concerning their interpretation (L. 240 – 241). The uncertainty on the conclusions ask for clarification.

Also the methods should be described and performed with more care to ease the

understanding of the manuscript. Here plastid transcript levels were standardized on actin-2. Although actin-2 is often used as reference gene, it is not a good choice in the presented study: Firstly, actin-2 transcript levels be affected by cold (see: eFP browser) and, secondly, actin-2 transcripts are nuclear encoded. Plastid genes respond with different dynamics than nuclear genes to various stresses. To analyze regulation that is independent from overall plastid effects, standardization should be performed on plastid genes. Due to the risk, the wrong choice of reference gene has, I recommend also to refer to two reference genes, test them for stability and spend more care on investigating the key aspect of the study.

(3) The conclusion are in the present form not sufficiently substantiated. Instead of "identified a mechanism" (end of the abstract), it is more "propose a hypothesis on a mechanism". The diurnal shift has not necessarily to be SIG5-dependent regulation, although SIG5 has earlier to be shown to take part in diurnal regulation. The analysis was performed at an early stage of plant development when the plastid cell division rates are high in the developing leaves. Cold could e.g. stopped cell division, plastid division or plastid maturation, which all decreases the transcript levels and can have especially strong effects on actin-standardized transcript levels. RbcL transcript data might be influenced by RbcS expression and RbcS protein import and could be in the cold more stabilized to prepare for increased osmolyte biosynthesis. In addition, the overall regulation of RNA synthesis and (specific) stability regulation could have impacted on the results, because cold has a strong impact on the expression of plastid targeting of RNA polymerases, splicing enzymes and RNA chaperons. For the conclusion on the relevance of SIG5, analysis of the effects in the mentioned SIG5-overexpression line (L. 233) could be very useful.

(4) There was no effect on the electrolyte leakage data and the freezing effects (FRZ / 20 °C) depicted in Fig. 4F could be the same in Col-0 and sig5-3. It could give us some information on cold regulation, which should be discussed. However, the experiments were performed with much older plants (in which cell division, chloroplast division and the photomorphogenetic sensitivity is much lower than in the seedlings used for other experiments, which should be considered in the discussion and interpretation of the data.

(5) The circadian transcript patterns show strong accumulation of Sig5 in the first half of the (suggested) night and a steady decrease during the (suggested) day. This is unusual for a blue-light entrained gene and different from the pattern described earlier (Belbin et al., 2017).

(6) The description of the RNA-Seq results is weak. Firstly, only gene numbers were provided for genes that respond in Col-0, but not in sig5-3, but not for genes responding in sig5-3, but not in Col-0. It is also not clear, how "do not respond" is defined. If the cut-of was set to $\log_2=1$, what about genes that responded 1.990 + 0.011 fold. How was "not responding" defined that it is clearly distinguishable from "respond"? As a reader, I would also be interested in the pattern of genes / biological processes that are differently regulated between Col-0 and sig5-3.

(7) The title should be specified. It is not informative in the present form.

(8) statistics:

(8a) Some results of the statistical analysis look strange, e.g. Fig. 1C Comparison of

SIG5 levels in hy5 at 19 °C and 4 °C. There are so strong differences that it is difficult to believe that the two experiments gave no significantly different results (more like this e.g. in Fig. 2C: W-A/C and M-A/C).

(8b) The numbers of biological replicates are very low with $n=2$ and $n=3$. For Fig. 4C-F, the pictures suggest effects, but the ranges observed overlap. With $n=2$, the in Fig. 4, it is not possible to decide if there is an effect or not, if the higher value of 2 in one line is higher than the lower level in the other line.

(8c) As the sample numbers differ strongly ($n= 2-60$), show standard deviation instead of standard error of mean.

(9) figure design: include labels above Fig 2C+F, 2D+G, 2E + H saying that they show data obtained in hy5, hyh and hy5 hyh, respectively. The legend could then be simplified to WS and mutant at 19 °C and 4°C (as in the description "statistical comparison").

(10) L. 80 – 82: Cold and drought / salt signaling share many components and drive overlapping signaling cascades. Wouldn't it be more interesting to study the osmotic / dehydration stress than just a single transcription factor.

(11) Logics behind (L. 74-75) "SIG5 transcripts accumulated in response to cold in the wild type but not in a hy5 hyh double mutant" in comparison to (L. 78 79) "HOWEVER, in darkness SIG5 transcripts were cold responsive and this was HY5/HYH-dependent".

(12) L. 79: "HY5/HYH" could be read as "HY5 or HYH" or "HY5 relative to / standardized on HYH".

(13) Provide information on the light qualities for all temperatures as the light quality changes in many light sources with temperature and light quality is a key issue in HY5-dependent gene regulation.

(14) Give key aspects of the experimental set-up in the results chapter that the reader can understand the experiments without going x-times into the methods description. Also give all important information in the manuscript. For example, for the cDNA synthesis in one of the mentioned references (Noordally et al., 2013) it is not appropriately described (e.g. information missing if the cDNA was oligo-dT, oligo-dTV or random primed) and in the other (Belbin et al., 2017) is not easy to access.

(15) Why were some plants grown in 6/16 and others in 12/12 light cycles?

(16) Nomenclature "ZT" is not explained in the main text and in the figure legends.

(17) L. 250 phrasing "SIG5 communicates": Does it communicate or is it affected?

(18) L. 258 phrasing "exerts exclusive control": That would be a strange mechanism. Check literature and rephrase accordingly.

(19) L. 280-282: Speculation, no experimental evidence shown

(20) L. 263 – 270: Speculation. There are several alternative explanations and, as

mentioned above, the impression on psbD regulation could result from inappropriate standardization.

- (21) Conclusion is "suggests" only, but is not really a conclusion.
- (22) l. 284: Word use "dissolve". It is not possible to dissolve salts in agar.
- (23) Citation missing for "Voom/Limma method" (L. 337)
- (24) Information on reference protein for Bradford assay is missing (L. 368)
- (25) Information on test criteria missing (e.g. hypergenomic test)

Reviewer #2:

Comments for the Author:

Low temperatures have widespread impacts on the physiology and development of plants. Plants evolve a variety of signaling mechanisms to respond to low temperature (LT). In the current paper, Cano-Ramirez et al report that the bZIP transcription factors ELOGATED HYPOCOTYL5 (HY5) and HY5 HOMOLOG (HYH) control the LT-induced expression of a nuclear-encoded sigma factor (SIGMA FACTOR5, SIG5). They further demonstrated that the expression of chloroplast psbD BLRP gene and even the transcriptome was altered in the sig5-3 mutant plants. Interestingly, the authors found that of the HY5/HYH-SIG5 pathway is related to the circadian clock, and regulates PSII D2 protein abundance and photosynthesis index, but not survival rate under LT conditions. Overall, these findings are interesting and provides a link between chloroplast signaling and nuclear response under LT conditions. Some specific comments are listed below.

1. Although there is a G-box cis-element presented on the promoter region of SIG5, the direct evidence showing the regulation of HY5/HYH on SIG5 promoter is missing, an effector-reporter assay may help to strengthen such interaction.
2. It seems the regulation of HY5/HYH-SIG5 exists at both light and dark conditions, while the LT-induced psbD BLRP only occurs in the light. Please discuss possible mechanism on this discrepancy.
3. Comparing Fig.1E to Fig.1D, the relative transcript level of psbD BLRP is quite different at ambient temperature, any comments on it?
4. Circadian gating of SIG5 expression at LT is also interesting in the paper. Is it possible to have a table listing the SIG5-dependent clock-associated genes?
5. Semi-quantitative capillary immunoassays were used in the current study. A reference is needed for this new technology. Further, it is not clear to the reviewer how many biological replicates were carried out in the study.

Reviewer #3:

Comments for the Author:

Nuclear encoded plastid sigma factors control the global switches of the plastid gene expression program. Sig5 is a unique plastid sigma factor that regulates transcription of the blue light-responsive promoter (BLRP) of psbD, which encodes the D2 protein of photosystem II. Previous studies have shown that SIG5 mediates low-temperature (LT) signaling to chloroplasts to promote the accumulation of psbD LRP transcripts.

Furthermore, the authors demonstrated in the previous paper that SIG5 is responsible for the circadian gating of light induction of psbD BLRP. However, the molecular mechanisms underlying SIG5-mediated activation of psbD LRP transcription in response to LT remain unclear. In addition, circadian gating of the response of SIG5 and psbD BLRP transcripts to LT has not been examined.

In this manuscript, the authors performed a detailed expression analysis of SIG5 and its target BLRP transcripts accumulation, revealing that the bZIP transcription factors HY5 and HYH play a critical role in the circadian gating of SIG5 and psbD LRP responses to LT. Furthermore, transcriptome analysis suggested the presence of an unexpected regulatory process mediated by SIG5 in the response of nuclear-encoded genes to LT. These studies are important for the understanding of the role of Sig5 in integrating LT and circadian signals. The paper is presented in a reasonably clear manner. However, much improvement can still be made to make it clearer and more concise. The reviewer felt that this manuscript needs some additional experiments to address the following concerns:

In Fig.2, the authors revealed that cold temperature caused the greatest accumulation of psbD BLRP during the subjective day, while cold-induced SIG5 transcript accumulation was most significant during the subjective night. This discrepancy suggests that the circadian gating of LT-induced psbD BLRP transcription might also be regulated by unidentified mechanisms in addition to the SIG5-mediated signaling (Fig5B). This additional signaling is partially dependent on HY5/HYH. However, the molecular mechanisms underlying this signaling have not been examined in detail in this study.

Previous studies demonstrate that global plastid transcription activity is arrested in the dark but activated by light (Link, 2003; Puthiyaveetil et al., 2010). It has been proposed that phosphorylated PEP and/or sigma factors tightly bind to promoters to arrest transcription under dark conditions. The HY5/HYH -dependent additional signaling might be involved in the regulation of light-dark oscillation of PEP activity which is required for the transcription of many photosynthesis genes, including psbD. I wonder whether the additional signal might regulate the transcription activity of PEP-dependent photosynthesis genes. Circadian analysis of cold-induced transcription activities of whole plastome-encoded genes (Run-on analysis) or circadian ChIP-seq analysis of PEP might shed light on the molecular mechanisms underlying the HY5/HYH -dependent additional signaling.

Since phospho-regulation of chloroplast transcription is regulated by chloroplast localized protein kinases, including cpK2, I suppose that it would also be helpful to examine the role of protein kinases in the HY5/HYH-dependent additional signaling.

Interestingly, this study revealed that SIG5 is required for the cold-responsive expression of some nuclear-encoded genes. As suggested by the authors, SIG5 may indirectly regulate the transcription of nuclear-encoded genes, probably via retrograde signals. Please examine the circadian gating of cold-induction of SIG5-dependent nuclear genes in some retrograde signal mutants to identify the retrograde signaling involved in the SIG5-dependent modulation of nuclear-encoded genes.

This study revealed that SIG5 is required to maintain photosynthetic efficiency and PSII D2 protein levels under long-term exposure to low-temperature conditions and freezing exposure. However, the reviewer felt this study is inconclusive since the response of SIG5 and psbD BLRP transcripts under long-term exposure to low-temperature conditions and freezing exposure is not examined. Photosynthetic efficiency decreases gradually under long-term exposure to low-temperature

conditions in WT plants. Is the abundance of PSII D2 protein also decreased gradually under low-temperature conditions? Furthermore, the authors did not examine the response of SIG5 and psbD BLRP transcripts to freezing. Please include these data to examine the possibility that Sig5-dependent maintaining of psbD BLRP transcription is responsible for maintaining the PSII D2 protein level and photosynthetic efficiency of PSII during prolonged cold and short-term freezing.

Minor comments

Please include the LT-induced expression data of HY5 and HYH transcripts under the experimental conditions used in this study.

p5 line 108, psbD BLRP was arrhythmic in hyh (Fig. S3B). correct?

p6 line146-p7 line 161, Some numbers of genes were not the same as those shown in the figures. e.g., p7 line 155, "959 transcripts that responded to cold at ZT29". However, the number of cold-responsive transcripts in Col-0 is 954 in Fig.3 and FigS4. Please check.

p11 line 263-270, I suppose that the AAAGTAAG sequence (AAG-box) in the psbD LRP is not a SIG5-binding site. Previous analyses demonstrated that AAG-binding factor (AGF) specifically binds to the AAG-box and interacts with PEP to promote transcription from the BLRP. Please edit the discussion.

Additional comments;

The abstract is written concisely and provides all necessary information. All figure legends properly provide the statistical processing and sample numbers. This manuscript also provides proper citations.

Link G. (2003) Redox regulation of chloroplast transcription. *Antioxid Redox Signal.* 5,79-87

Puthiyaveetil S, Ibrahim IM, Jelčić B, Tomasić A, Fulgosi H, Allen JF. (2010) Transcriptional control of photosynthesis genes: the evolutionarily conserved regulatory mechanism in plastid genome function. *Genome Biol Evol.* 2, 888-96

Author Rebuttal to Initial comments

Cano-Ramirez, Panter et al. –

Response to Reviewers Reviewer

#1:

Concerns:

(1) The manuscript describes its results without careful consideration of the knowledge of our time. It hardly reflects the literature. For example, the statement “ we know relatively little about how information concerning low temperature conditions is communicated to chloroplasts” (abstract) is simply wrong. We know a lot. One of the best studied examples are cold regulation of the expression of the chloroplast membrane protein Cor15A (external sensing) and cold regulation of photosynthesis (internal sensing). We also know that SIG5 has (like almost all genes) a complex promoter with binding sites for various transcriptional regulators (= more than HY5, HYH and ATHB17) and responds to various abiotic and biotic stress treatments (see e.g. eFP browser). The introduction is in the present form very, very short and lacks lot of information the reader requires to understand the experiments and the results.

Response: We apologise that the introduction was insufficiently comprehensive. The previous manuscript version aimed to restrict the word count of the introduction, but we realize this was inappropriate. We revised our paper to expand the introduction (lines 43-81), which now covers a broader range of literature. Some specific details from the literature that are necessary to understand specific experiments are also mentioned briefly in the Results. We thank the reviewer for drawing our attention to this problem.

(2) HY5 and HYH control photomorphogenesis. In many photomorphogenic and signaling processes, lack of HY5 can cause the phenotype on its own. Here, lack of HY5 and lack of HYH, both, had on their own no cold effect on SIG5. Cold-induction of SIG5 was only disturbed in *hy5 hyh* double mutants. Such combinatory effect is unusual and should have been discussed in more detail.

Response: We welcome the reviewer’s comment about roles for HY5 alone, and HY5/HYH combinatorial effects. In response to comments from this reviewer and other reviewers, we conducted additional statistical analyses (Fig. 1, Fig. 2, Fig. S3) and conclude that the *hy5* mutant alone disturbs the expression of *SIG5* under control temperature conditions. This

10includes alterations in the circadian rhythm of *SIG5*. In comparison, the *hy5 hyh* double mutant is required to abolish the response of *SIG5* to cold (Fig. 1B). This differs from our previous interpretation that the *hy5* single mutant was without effect. We revised both the Results and Discussion to reflect this different and more accurate interpretation. Key additional Results text concerning this are, “The peak transcript abundance of *SIG5* was reduced significantly at a subset of timepoints in the *hy5* or *hyh* single mutants, and at a greater number of timepoints in the *hy5 hyh* double mutant (Fig. S3A). The peak abundance of *psbD* BLRP reduced significantly at some timepoints in the *hy5* mutant and *hy5 hyh* double mutant, but not in the *hyh* single mutant (Fig. S3B),” (lines 131-136), and “The amplitude was reduced more in *hy5* compared with *hyh*, and was comparable between *hy5* and *hy5 hyh* (Fig. S3A, B; Dataset S1),” (lines 142-143) and “It appears that the *hy5* single mutant affects *SIG5* and *psbD* BLRP transcript accumulation at control temperatures, whereas the *hy5 hyh* mutant is required to abolish its response to cold” (189-191). We appreciate the reviewer’s request to check this further.

Photomorphogenesis has also a strong impact on chloroplast development and might feed-back on *SIG5*. Such an effect can cause un- (less) specific regulation, such as the described co-regulation of *psaC* and *psbD* and asks for more careful experiment design. In the discussion, the authors themselves present doubts concerning their interpretation (L. 240 – 241). The uncertainty on the conclusions ask for clarification.

Response: We thank the reviewer for their comment about careful experimental design and interpretation. In response to comments from several reviewers about roles for HY5 in the regulation of this pathway, we have added additional experimentation (including investigation of SIG5 promoter regulation by HY5, and examination of HY5 transcript levels), and revised the analysis and interpretation (new statistical analysis of data, removal of over-interpretation from Discussion). We believe these changes clarify the interpretation of the results. We consider it appropriate and balanced to describe alternative interpretations in the Discussion in cases where there could be more than one explanation for a process, and where it opens questions for future experiments.

Also the methods should be described and performed with more care to ease the understanding of the manuscript. Here plastid transcript levels were standardized on actin-2. Although actin-2 is often used as reference gene, it is not a good choice in the presented study: Firstly, actin-2 transcript levels be affected by cold (see: eFP browser) and, secondly, actin-2 transcripts are nuclear encoded. Plastid genes respond with different dynamics than nuclear genes to various stresses. To analyze regulation that is independent from overall plastid effects, standardization should be performed on plastid genes. Due to the risk, the wrong choice of reference gene has, I recommend also to refer to two reference genes, test them for stability and spend more care on investigating the key aspect of the study.

Response: We welcome the comments from the reviewer about reference gene choice. The eFP browser indicates that *ACTIN2* transcript accumulation is altered slightly by longer-term cold treatments, rather than short (e.g. 3 h) cold treatments. However, we cannot exclude the possibility that under our experimental conditions, *ACT2* might respond to cold. To control for this, we repeated key experiments using two additional reference genes (*UBQ10*, *EF-1A* that do not appear to respond to cold at all). Using these reference genes, we found that *SIG5* transcript levels increase in response to cold, *SIG5* expression varied according to the time of day, and its response to cold was altered in the *hy5 hyh* double mutant. These conclusions are the same as when *ACT2* was used as a reference gene. We also noticed that the average CT value of the *ACT2* PCR reactions was unaltered by cold, suggesting *ACT2* transcript abundance was unaltered by our cold treatment (we did not include these data). Finally, we do not think that there is an overall change in the abundance of chloroplast transcripts in response to cold- as could happen if the number of chloroplasts changed, or there is a change in DNA topology- because there is no systematic change in chloroplast transcript abundance in our RNA sequencing analysis (Table S2). Furthermore, another study suggests that chloroplast transcript abundance does not change systematically in response to cold (Gao et al. Plant Cell 2022). We revised our manuscript to add these new data (Fig. S8D), and revised the Methods to explain all of these considerations (lines 421-430). We thank the reviewer for questioning our approach because it motivated us to double-check the validity of our methods, and improve the paper.

(3) The conclusion are in the present form not sufficiently substantiated. Instead of “identified a mechanism” (end of the abstract), it is more “propose a hypothesis on a mechanism”. The diurnal shift has not necessarily to be SIG5-dependent regulation, although SIG5 has earlier to be shown to take part in diurnal regulation. The analysis was performed at an early stage of plant development

when the plastid cell division rates are high in the developing leaves. Cold could e.g. stopped cell division, plastid division or plastid maturation, which all decreases the transcript levels and can have especially strong effects on actin-standardized transcript levels. RbcL transcript data might be influenced by RbcS expression and RbcS protein import and could be in the cold more stabilized to prepare for increased osmolyte biosynthesis. In addition, the overall regulation of RNA synthesis and (specific) stability regulation could have impacted on the results, because cold has a strong impact on the expression of plastid targeting of RNA polymerases, splicing enzymesand RNA chaperons. For the conclusion on the relevance of SIG5, analysis of the effects in the mentioned SIG5-overexpression line (L. 233) could be very useful.

Response: We agree that “identified a mechanism” was a poor word choice, because it implies knowledge of every regulatory step. We think that individual publications rarely identify every aspect of a mechanism. We revised the Abstract extensively to make this more accurate. On the point about actin normalization, we appreciate the reviewer’s concern and refer them to our response above.

We agree that plants overexpressing SIG5 from the plastid genome would be useful for this work. We previously tried to obtain the seeds from the Shiina lab, and co-author Kan Tanaka has tried to do so independently. Unfortunately, the plants have not been made available to us so instead we highlighted this as an interesting area for the future in our Discussion (lines 319-322). We do not think it would be informative to examine nuclear-encoded 35S::SIG5 plants. In our hands, 35S::SIG5 reduces the amplitude of circadian rhythms of *psbD* BLRP transcript accumulation, but does not upregulate *psbD* BLRP transcript levels.

(4) There was no effect on the electrolyte leakage data and the freezing effects (FRZ / 20 °C) depicted in Fig. 4F could be the same in Col-0 and *sig5-3*. It could give us some information on cold regulation, which should be discussed. However, the experiments were performed with much older plants (in which cell division, chloroplast division and the photomorphogenetic sensitivity is much lower than in the seedlings used for other experiments, which should be considered in the discussion and interpretation of the data.

Response: The reason why we conducted the electrolyte leakage assay with older plants is because it is necessary to carefully control the quantity and consistency of tissue used for each assay, otherwise the data are rather noisy. This level of control is only possible with larger leaves. To allow a comparison with younger plants, we performed a visual comparison of freezing damage in 14-day old seedlings (Fig. S7B, C). As with older plants (Fig. S7D, E), the *sig5* mutation had no effect upon freezing tolerance of the 14-day old seedlings. The reviewer’s comment made us realise that we did not discuss this properly, including our choices of developmental stages. Therefore, we revised the Results to explain, “We conducted electrolyte leakage analysis on mature leaves rather than younger plants to enable the very consistent sampling of leaf discs that is necessary to limit data noise. Nevertheless, freezing tolerance was unaltered by the *sig5-3* mutant, relative to the wild type, in both 14 day old seedlings and mature plants (Fig. S7C-F)” (lines 295-299). We believe this clarifies for readers (a) why we chose two developmental stages, and (b) the freezing tolerance results are similar at both developmental stages. We thank the reviewer for asking us to justify our experimental design.

(5) The circadian transcript patterns show strong accumulation of Sig5 in the first half of the (suggested) night and a steady decrease during the (suggested) day. This is unusual for a blue-light entrained gene and different from the pattern described earlier (Belbin et al., 2017).

Response: The reviewer noticed a difference between the 24 h rhythm of accumulation of

SIG5 transcripts in this study and in another paper (Belbin et al. *New Phytol.* 2017). We believe this is due to differences in the light conditions. The data in this study (e.g Fig. 2A, Fig. S3) are exactly consistent with our previous study (Noordally et al. *Science* 2013), where *SIG5* transcripts start to accumulate during the subjective night, peak around subjective dawn, and then decrease. The experiments were conducted under exactly the same light conditions and growth chamber type as Noordally et al. 2013 (90 $\mu\text{mol m}^{-2} \text{s}^{-1}$ of white light). The experiments in Belbin et al. 2017 occurred under lower intensity monochromatic light of specific wavelengths, using LED panels (25 $\mu\text{mol m}^{-2} \text{s}^{-1}$).We realise that this difference might puzzle readers, so we resolved this through two revisions. First, we explained this difference in the Results (“Under these control temperature conditions, we compared the circadian rhythms of *SIG5* and *psbD* BLRP transcript accumulation in the *hy5*, *hyh*, and *hy5 hyh* mutants with the wild type. *SIG5* transcript abundance increased during the subjective night to reach a peak around subjective dawn (Fig. S3A), which is consistent with other studies conducted under constant white light conditions of 90 $\mu\text{mol m}^{-2} \text{s}^{-1}$ [24]. This contrasts the dynamics of *SIG5* transcript abundance under monochromatic light, where it peaks later in the subjective day [26],” lines 125-131). Second, we added the light spectra as Fig. S8A-C, because we did not previously include this information. We refer to these spectra in the Methods (lines 398-400). We thank the reviewer for asking us to look closer at this difference.

(6) The description of the RNA-Seq results is weak. Firstly, only gene numbers were provided for genes that respond in Col-0, but not in *sig5-3*, but not for genes responding in *sig5-3*, but not in Col-0. It is also not clear, how “do not respond” is defined. If the cut-of was set to $\log_2=1$, what about genes that responded $1.990 + 0.011$ fold. How was “not responding” defined that it is clearly distinguishable from “respond”? As a reader, I would also be interested in the pattern of genes / biological processes that are differently regulated between Col-0 and *sig5-3*.

Response: The lists of genes responding to cold in *sig5-3*, but not in Col-0, was provided in Supplemental Dataset S4 of our previous submission. This was also summarized on lines 219-222. We apologise that the location of these data was unclear. To correct this, we revised lines 219-222 to emphasise the location of these data.

The reviewer also asked about the pattern of genes that are differentially regulated between Col-0 and *sig5-3*. The sets of genes that are differentially regulated between Col-0 and *sig5-3* (at two timepoints) was provided in Dataset S2 of our previous submission. We agree with the reviewer that these gene sets are interesting, but did not discuss them in our previous submission. Therefore, we revised the Results to include this information, and emphasize the location of the gene lists (“Under control temperature conditions, a relatively small number of transcripts were differentially expressed between Col-0 and *sig5-3* at the timepoints examined (29 and 42 transcripts differentially expressed at ZT29 and ZT45, respectively), with no significant GO-term enrichments within these gene sets (Dataset S2).” Lines 198-201). We thank the reviewer for noticing this omission.

Concerning transcripts that do not respond to cold, the reviewer will be aware that it is necessary to select statistical thresholds when conducting RNAseq analysis, and adhere to those thresholds consistently. In this case, we selected a combination of fold change $> \log_2$ and $p \leq 0.01$, which we consider to be stringent. Some transcripts will respond, at a level that falls below the statistical threshold. We understand the reviewer’s concern, and make a number of revisions to improve this. First, we revised the manuscript in several places to be specific about the definitions of transcripts that do and do not respond to cold, so we revised the manuscript to provide specific information about this (lines 191-193; 203-206). Second, our previous submission was ambiguous around the idea of “different magnitudes of cold-responsiveness” in the wild type and mutant, when we meant “significantly cold-responsive”

or “not significantly cold-responsive.” To resolve this, we checked the precision of this Results section and revised several sentences (“Together, this indicates that some transcripts required SIG5 to respond significantly to the cold treatment.” Lines 223-224; “...we hypothesized that the set of transcripts that responded significantly to the cold treatment in the wild type but not *sig5-3* mutant might be enriched with circadian-regulated transcripts.” Lines 225-227; “...the set of transcripts that responded significantly to the cold treatment in Col-0 but not in *sig5-3*...” lines 229-230; “...responded significantly to the cold treatment in *sig5-3* but not in the wild type...” lines 232; “...whether the set of transcripts that responded significantly to cold in only Col-0 or *sig5-3*...” (lines 241-243). Finally, weremoved a speculative sentence at the end of this section that made claims about new regulatory processes. We believe these revisions have improved the precision and specificity of this section.

(7) The title should be specified. It is not informative in the present form.

Response: Reviewer 1 thought the title could be more specific. We tend to agree. It can be difficult to balance general interest and scientific detail. Following the reviewer's advice, we revised the title to, "Low temperature and circadian signals converge upon the sigma factor SIG5, which confers photosynthetic resilience to long-term cold."

(8) statistics:

(8a) Some results of the statistical analysis look strange, e.g. Fig. 1C Comparison of SIG5 levels in *hy5* at 19 °C and 4 °C. There are so strong differences that it is difficult to believe that the two experiments gave no significantly different results (more like this e.g. in Fig. 2C: W-A/C and M-A/C).

Response: In response to the reviewer's comment, we reran all statistical analyses on Fig. 1 and Fig. 2, using two separate packages (R and SPSS). We identified a number of problems with the original analysis, and also decided that the previous design of Fig. 2 and S3A/B was confusing. The corrected statistical analysis is shown in our newly submitted versions of Fig. 1 and Fig. 2. For Fig. 2C-H, we reduced the number of comparison types within the figure, comparing only the wild type and mutant at each temperature in the main figure. We also revised the figure to clarify the labelling and colours.

This reanalysis produced a change in interpretation, because it identified that the *hy5* mutant alone affects the accumulation of *SIG5* and *psbD* BLRP transcripts under control temperature conditions, at certain times of day (Fig. S3A, B). It supports the reviewer's view that *hy5* mutants alone can give a phenotype. We found that the *hyh* mutant alone did not consistently alter *SIG5* or *psbD* BLRP accumulation compared with the wild type (Fig. 1B, Fig. 2D). The *hy5 hyh* double mutant strongly attenuated *SIG5* expression under cold temperature conditions, and always affected the accumulation of *psbD* BLRP transcripts under cold temperature conditions. It seems that the cold-responsive phenotype is stronger in the double mutant. To incorporate this altered interpretation, we made changes throughout the Results section from line 120-161 and 168-191. This included revision of the entire Results section concerning Fig. 2 and Fig. S3A, B, to explain that the dynamics of *SIG5* in *hy5* single mutant compared with the *hy5 hyh* double mutant. We thank the reviewer for asking us to check our statistical analysis because this resulted in a correction to the data interpretation.

(8b) The numbers of biological replicates are very low with $n=2$ and $n=3$. For Fig. 4C-F, the pictures suggest effects, but the ranges observed overlap. With $n=2$, the in Fig. 4, it is not possible to decide if there is an effect or not, if the higher value of 2 in one line is higher than the lower level in the other line.

Response: We appreciate the comment from the reviewer, who will be aware that protein blots can sometimes be variable. In response to their comment and a comment from another reviewer, we revised the figure to make clear that the experiment included two independent biological replicates (indicated by circles on Fig. 4D and Fig. 4F). The abundance of both D2 and PSI-C is reduced consistently in *sig5-3* mutant after freezing, which is the conclusion drawn within the Results (“Both PSII D2 and PSA-C protein abundance was decreased consistently in *sig5-3* plants after this freezing treatment, compared to Col-0 under control temperature conditions.”, lines 268-270). We removed a statement about statistical significance, which was inappropriate. We think it unlikely that further replication wouldchange the conclusions (such an approach might be considered to be p-value hacking).

(8c) As the sample numbers differ strongly ($n = 2-60$), show standard deviation instead of standard error of mean.

Response: The variation in sample sizes occurs because different sample sizes were used for different types of experiments. We selected sample sizes that are appropriate for each type of experiment, based on our previous experience, the scale of the experiment, and type of data analysis required. It is never our intention to quantitatively compare experiments where (for example) $n = 3$ (RT-qPCR) and $n = 60$ (physiological assays). We checked the figure legends to ensure that the sample size used for each type of experiment was explained clearly, and also revised the legends of Fig. S3 and Fig. S7 to ensure the sample sizes are explained properly.

(9) figure design: include labels above Fig 2C+F, 2D+G, 2E + H saying that they show data obtained in hy5, hyh and hy5 hyh, respectively. The legend could then be simplified to WS and mutant at 19 °C and 4°C (as in the description “statistical comparison”).

Response: We welcome this suggestion, and decided to simplify the labelling of the entire figure. This included changes the presentation of statistical comparisons (see our earlier response), which we believe improves its clarity. We decided to retain the colours used for the mutant lines, to provide colour consistency with other figures. We restructured the legend of Fig. 2 to improve its logic (discusses A, B, then C-H), and explain the statistical comparisons clearly. We thank the reviewer for motivating us to improve the clarity of this figure.

(10) L. 80 – 82: Cold and drought / salt signaling share many components and drive overlapping signaling cascades. Wouldn't it be more interesting to study the osmotic / dehydration stress than just a single transcription factor.

Response: We agree that SIG5-mediated gene regulation participates in responses to other types of abiotic stresses. However, this was not the focus of our study because such stresses have been examined elsewhere in the context of SIG5 (e.g. Nagashima et al. PCP 2004; Zhao et al. Sci. Rep. 2017). We think that investigating additional types of stresses is beyond the scope of our focus here on cold temperature responses.

The reviewer commented specifically upon the text on lines 101-105 about ATHB17. We double-checked that this was not misleading. We decided that this needed revision, so revised it to (“It is known that SIG5 transcript accumulation in response to salinity involves HOMEBOX-LEUCINE ZIPPER PROTEIN17 (ATHB17) [27]. We found that cold induction of SIG5 in the light was not altered significantly in *athb17* mutants (Fig. S2B), suggesting that ATHB17 does not participate in this response to cold.” (lines 101-105).

(11) Logics behind (L. 74-75) “SIG5 transcripts accumulated in response to cold in the wild type but not in a hy5 hyh double mutant” in comparison to (L. 78 79) “HOWEVER, in darkness SIG5

transcripts were cold responsive and this was HY5/HYH-dependent”.

Response: The reviewer commented on an inconsistency concerning the *SIG5* transcript expression phenotype of the *hy5 hyh* double mutant. We agree that this was inconsistent and corrected the text concerning dark conditions to, “However, in darkness *SIG5* transcripts were cold-responsive in the wild type, but not in the *hy5 hyh* double mutant (Fig. 1C).”; lines 99-101. This change ensures logical consistency between the light and dark conditions. We thank the reviewer for this suggestion.(12) L. 79: “HY5/HYH” could be read as “HY5 or HYH” or “HY5 relative to / standardized on HYH”.

Response: We agree that HY5/HYH was ambiguous at this location and elsewhere. We followed the reviewer’s advice and revised this throughout the paper to be specific.

(13) Provide information on the light qualities for all temperatures as the light quality changes in many light sources with temperature and light quality is a key issue in HY5-dependent gene regulation.

Response: We agree with the reviewer that the temperature can change the output of some light sources. Therefore, we added the spectral fluence at control and cold temperatures to supplemental information (Fig. S8A, B). We revised the Methods to explain these data (“The light spectrum was similar when the chambers were set to control and cold temperature conditions (Fig. S8A, B; Li-cor LI-180 spectrometer).” Lines 398-400). Fortunately, in our growth chambers the light sources are physically isolated from the growing space, so the temperature has little effect upon the light output or spectrum.

(14) Give key aspects of the experimental set-up in the results chapter that the reader can understand the experiments without going x-times into the methods description. Also give all important information in the manuscript. For example, for the cDNA synthesis in one of the mentioned references (Noordally et al., 2013) it is not appropriately described (e.g. information missing if the cDNA was oligo-dT, oligo-dTV or random primed) and in the other (Belbin et al., 2017) is not easy to access.

Response: We apologise that the reviewer had to swap repeatedly between the Results, Methods and other publications to locate key methods details. We have now added to the Methods section full detail of the methods associated with RT-qPCR (lines 415-418). We also noticed that the electrolyte leakage (freezing tolerance) assay was not described properly, so revised the manuscript to explain this fully (lines 504-527).

We are not exactly sure which methods the reviewer wished to see in the Results section. Therefore, we checked the entire Results for cases where brief description of methods might help the narrative. We made many small changes to improve this (e.g. describing, where relevant, the duration of cold treatments). We also made revisions to explain clearly the structure of circadian timecourse experiments, e.g. “We investigated this by cultivating seedlings for 11 days under cycles of 12 h light and 12 h darkness, and then transferring the seedlings to conditions of constant light and temperature to monitor the free-running rhythm of transcript abundance.” (lines 122-125) and “To evaluate further the contribution of HY5 and HYH to circadian rhythms of *SIG5* and *psbD* BLRP transcript accumulation, we compared the amplitude of these rhythms in *hy5*, *hyh*, *hy5 hyh* and the wild type, using the MetaCycle circadian rhythm analysis software [34].” (lines 137-139). We hope that these and other additions make the Results easier to comprehend.

(15) Why were some plants grown in 6/16 and others in 12/12 light cycles?

Response: The reviewer asked why some plants were grown in 16/8 cycles, whilst others (mainly) in 12/12 cycles. We used 16/8 cycles for one experiment because we wanted this particular experiment to be directly comparable to a previous paper about HY5 and cold responses (Catala et al. 2011). We apologise that we did not explain this in our paper. We revised the Methods to explain this (“For this experiment, 16 h days were used to increase similarity of experimental design with a previous study on HY5 and low temperature responses [30].” (lines 397-398). By default, we use 12/12 cycles because this produces a more symmetrical waveform in timecourse experiments, which is readily analysed with tools such as MetaCycle.(16) Nomenclature “ZT” is not explained in the main text and in the figure legends.

Response: The reviewer noticed that unfortunately we did not define ZT. To address this, we revised the results to include an explanation and definition upon first use (“zeitgeber time (ZT) 41; i.e. 41 h after the final dawn under constant free-running conditions”, lines 172-173). We also added an explanation of ZT to the legends of Fig. 3 and Fig. S4, because it is relevant to those figures. We tried to minimise the use of chronobiology jargon to make the paper more widely accessible, but in a few places such as this we feel that it is necessary.

(17) L. 250 phrasing “SIG5 communicates”: Does it communicate or is it affected?

Response: We apologise that this sentence was insufficiently precise. To correct this, we revised it to, “SIG5 is required for circadian regulation of a set of chloroplast transcripts.” (line 351). We had tried to keep the Discussion quite short, but we realize it lacked nuance, so appreciate this comment.

(18) L. 258 phrasing “exerts exclusive control”: That would be a strange mechanism. Check literature and rephrase accordingly.

Response: We agree that this description was poor. We revised the interpretation to say, “Therefore, multiple circadian clock-related factors appear to converge upon the promoter of SIG5, with HY5 and HYH representing one of these mechanisms.” (line 358-360).

(19) L. 280-282: Speculation, no experimental evidence shown

Response: We apologise for the poor wording of this conclusion section of the Discussion. This concerned the involvement of sigma factors in responses to cold temperatures in various kingdoms of life. In response to the reviewer’s comment, we rewrote this to improve its accuracy (“Sigma factors allow bacteria and cyanobacteria to respond to cold temperature conditions [80-83]. Our experiments identify that sigma factors also participate in responses to cold temperatures in plants. Therefore, taken together with studies in bacteria and cyanobacteria [80-83], it appears that sigma factors are involved cold temperature responses in both prokaryotes and eukaryotes.”; lines 369-373). We apologise for the previous over-speculation.

(20) L. 263 – 270: Speculation. There are several alternative explanations and, as mentioned above, the impression on psbD regulation could result from inappropriate standardization.

Response: We agree that this section was speculative. It interested us at the time of writing but in response to this comment- and a comment of another reviewer- we decided to remove this. We condensed this to a comment about effects of the *sig5-3* mutation on the nuclear-encoded transcriptome. This section of the Discussion did not concern *psbD* expression. We rewrote this section as, “It is interesting that a set of nuclear-encoded transcripts are cold-responsive in the wild type, but not in *sig5-3*. This phenotype of the *sig5-3* mutant suggests

indicates that a function of SIG5 can influence nuclear-encoded gene expression. We speculate that this is likely to occur indirectly, perhaps through metabolic alterations arising from altered chloroplast function in the *sig5-3* mutant.”(lines 363-367). We indicate clearly which part of this revised text is speculation. The focus of our study is not retrograde signalling, so we do not wish to speculate further about mechanisms that cause this phenotype.

(21) Conclusion is “suggests” only, but is not really a conclusion.Response: We welcome the reviewers' comment. We tend to be cautious about claims of absolute discovery and novelty, because it is rarely possible to "prove" something with complete confidence in science. In response to the reviewer's comment, we revised the wording to, "Our experiments identify a new regulator of cold temperature responses of chloroplasts, and establish that a sigma factor contributes to protection of photosynthesis before and during freezing." (lines 373-375).

(22) l. 284: Word use "dissolve". It is not possible to dissolve salts in agar.

Response: We agree that this was a poor choice of words, and corrected this (line 383).

(23) Citation missing for "Voom/Limma method" (L. 441)

Response: We added a citation of the original article for the Voom/Limma method (line 455).

(24) Information on reference protein for Bradford assay is missing (L. 368)

Response: We apologise this detail was missing, and have added this to the Methods ("Sigma reagent B6916 for protein range 0 – 1.4 mg/mL; calibrated across protein concentration range 0 - 1.2 mg/mL using a BSA standard"), line 494-496).

(25) Information on test criteria missing (e.g. hypergeometric test)

Response: We apologise that the hypergeometric test was not explained clearly. We are unsure about which test statistics the reviewer considered to be missing, because the statistics are provided in Fig. S4. However, we noticed that we did not explain properly how the test was conducted. To correct this, we revised the Methods to describe the test and its rationale ("We determined whether statistically significant overlaps existed between sets of transcripts by using a hypergeometric test, which considers whether the overlap between two sets of genes is significantly different from the size of an overlap arising from two randomly-drawn sets of genes. This involves calculation of the representation factor, which is the actual number of genes in the intersection, divided by the expected number of genes in the intersection; thus a value > 1 indicates a greater number of genes than expected, and a value < 1 indicated fewer genes than expected). The probability of this intersection occurring was calculated using a normal approximation of the exact hypergeometric probability [93, 94]." Lines 458-466). Furthermore, we revised the legend of Fig. S4 to refer the reader to this methods section. We believe this provides clarification.

Reviewer #2:

Comments for the Author:

Low temperatures have widespread impacts on the physiology and development of plants. Plants evolve a variety of signaling mechanisms to respond to low temperature (LT). In the current paper, Cano-Ramirez et al report that the bZIP transcription factors ELOGATED HYPOCOTYL5 (HY5) and HY5 HOMOLOG (HYH) control the LT-induced expression of a nuclear-encoded sigma factor (SIGMA FACTOR5, SIG5). They further demonstrated that the expression of chloroplast psbD BLRP gene and even the transcriptome was altered in the sig5-3 mutant plants. Interestingly, the authors found that of the HY5/HYH-SIG5 pathway is related to the circadian clock, and regulates PSII D2 protein abundance and photosynthesis index, but not survival rate under LT conditions. Overall, these findings are interesting and provides a link between chloroplast signaling and nuclear response under LT conditions. Some specific comments are listed below.

We are pleased that Reviewer 2 considers our findings to be interesting, and found their suggestions very helpful.

1. Although there is a G-box cis-element presented on the promoter region of SIG5, the direct evidence showing the regulation of HY5/HYH on SIG5 promoter is missing, an effector-reporter assay may help to strengthen such interaction.

Response: We agree it would be useful to know whether HY5 or HYH directly regulates the SIG5 promoter. In response to the reviewer's suggestion, we tested this experimentally. It did not prove as straightforward as we had anticipated. Using transient expression in *Nicotiana benthamiana*, with 35S::HY5 or 35S::HYH as an effector and SIG5::LUCIFERASE as a reporter, we observed downregulation of SIG5::LUCIFERASE in the presence of these transcription factors (see graphs below; normalized to a transformation efficiency control). This puzzled us because it is not consistent with the prevailing data in the literature, including HY5 ChIP-seq from the Xing Wang Deng lab, our data, and data from the Kan Tanaka lab. We reason that this downregulation might be a *N. benthamiana*-specific phenomenon, a consequence of unnaturally high levels of HY5 or HYH (e.g. transcriptional squelching), or there are aspects of the regulation of this promoter that are yet to be discovered. We did not include these data in the paper because they could be artefacts, so instead performed further experiments to test this.

As an additional strategy, we bombarded particles carrying SIG5::LUCIFERASE into wild type, *hy5* and *hyh* mutant *Arabidopsis* plants, and measured luciferase bioluminescence at dawn and at dusk. This used protocols similar to those developed by the Tokitaka Oyama lab (Muranaka and Oyama 2016, Science Advances). In this experiment, SIG5::LUCIFERASE activity was reduced in the mutants compared with the wild type. We included these data (Fig. S3C) because we think this is an effective experiment: the results agree with the published literature, loss of function experiments are more revealing than overexpression experiments (because they reveal necessity rather than sufficiency), and the experiment will lack off target or transcriptional squelching issues that might occur with *N. benthamiana*. We describe these finding on lines 148-153.

The different results from these two experiments makes us cautious about claiming that HY5 and/or HYH *directly and positively* regulates the SIG5 promoter. Experiments measuring transcript levels (Fig. 2C) and bombarded SIG5::LUC activity (Fig. S3C) support the notion

28

that HY5 and HYH *regulate pSIG5* activity, but it does not prove they activate *pSIG5*. Therefore, in addition to these new data, we checked the manuscript throughout to avoid claims about direct activation.

We thank the reviewer for their suggestion, it was a great idea. Data collection took longer than expected, but the new results have provided the precision, nuance and depth of our study.

2. It seems the regulation of HY5/HYH-SIG5 exists at both light and dark conditions, while

the LT-induced *psbD* BLRP only occurs in the light. Please discuss possible mechanism on this discrepancy.

Response: The reviewer is correct that *psbD* BLRP responds only to LT in the light, whereas *SIG5* responds to LT in both light and dark. We agree that this difference is worth discussing. Our interpretation is that in darkness, LT upregulates *SIG5* transcript levels, which might increase *SIG5* protein levels within chloroplasts. However, light is required for the association of PEP with chloroplast DNA and for PEP assembly, with a potential role also for redox regulation. Therefore, one possibility is that upregulation by cold of *SIG5* does not alter *psbD* BLRP transcription in darkness because PEP is not active. Another possibility is that in darkness, *SIG5* is not imported efficiently into chloroplasts and thus does not reach a threshold required to generate *psbD* BLRP transcripts. To address this, we added an entire new section of the Discussion that includes this interpretation (lines 311-324). This also addresses some questions from another reviewer. We thank the reviewer for their helpful suggestion.

3. Comparing Fig.1E to Fig.1D, the relative transcript level of *psbD* BLRP is quite different at ambient temperature, any comments on it?

Response: The reviewer asked why *psbD* BLRP relative transcript abundance differs between Fig. 1D and Fig. 1E. These experiments were conducted in different background accessions, and at different times, so the normalization will be different. We described in the Results a potential effect of the Arabidopsis accession upon the responses (Col-0 vs Ws) (lines 186-189).

4. Circadian gating of *SIG5* expression at LT is also interesting in the paper. Is it possible to have a table listing the *SIG5*-dependent clock-associated genes?

Response: Reviewer 2 wished to know whether there are *SIG5*-dependent circadian clock-associated genes. Under control temperature conditions, no circadian clock transcripts were altered by the *sig5* mutation (Dataset S2). This is consistent with the observation that circadian clock function is unaltered by *sig5* mutants under control temperature conditions (Noordally et al. 2013). In response to a cold treatment, the putative clock-associated transcript *LNK4* responded significantly to cold in the wild type but not the *sig5* mutant. Furthermore, *BBX19* was upregulated by cold in the *sig5* mutant, but not in the wild type. This is a small number of genes, so it would be inefficient to add a table. Instead, we added a section to the Results to clarify which clock genes have altered cold responses in the *sig5* mutant (“The only circadian clock-associated transcript that was significantly cold-induced in Col-0 but not *sig5-3* was *NIGHT LIGHT-INDUCIBLE AND CLOCK-REGULATED4 (LNK4)* (at ZT29; Dataset S4), although the role of *LNK4* within circadian regulation remains uncertain [51, 52]. Furthermore, transcripts encoding the zinc finger protein B-BOX DOMAIN PROTEIN19 (*BBX19*) were upregulated by cold in *sig5-3*, but not the wild type (Dataset S4). *BBX19* is thought to repress the promoters of certain morning-phased circadian clock components [53].” Lines 231-238). We thank the reviewer for this good suggestion.

5. Semi-quantitative capillary immunoassays were used in the current study. A reference is needed for this new technology. Further, it is not clear to the reviewer how many biological replicates were carried out in the study.

Response: Reviewer 2 requested inclusion of references for the ProteinSimple semi-quantitative automated capillary immunoassay system. We agree this is useful, because this technology is unfamiliar to some readers. Its use does not seem widespread (yet) in plant sciences. We revised the Methods to cite two examples of recent uses of this method, and the original study that describes the method (Lück et al. 2021). Within our institute (John Innes Centre), the method seems to have much better reproducibility between technical and

biological replicates compared with conventional Western blotting. Perhaps its use will become more widespread in future.

We apologise that the replication was not explained clearly. Replication comprised two completely independent experiments per treatment/genotype, with three technical repeats within each experiment. We revised the style of the graphs in Fig. 4D and F to make this clear, because the previous data representation was misleading. We also revised the figure legend to include detail of replication (“Circles on plots indicate result from each independent experiment (two independent repeats, each with three technical replicates).” Lines 594-596). We show representative examples of the gels (Fig. 4C, E), with the others in supplemental material.

Reviewer #3:

Comments for the Author:

In Fig.2, the authors revealed that cold temperature caused the greatest accumulation of *psbD* BLRP during the subjective day, while cold-induced *SIG5* transcript accumulation was most significant during the subjective night. This discrepancy suggests that the circadian gating of LT-induced *psbD* BLRP transcription might also be regulated by unidentified mechanisms in addition to the *SIG5*-mediated signaling (Fig5B). This additional signaling is partially dependent on *HY5/HYH*. However, the molecular mechanisms underlying this signaling have not been examined in detail in this study.

Response: The reviewer asked about different phases of the response to cold of *SIG5* and *psbD* BLRP. Later phase of *psbD* BLRP relative to *SIG5* occurs also under control temperature conditions, when the phase of *SIG5* is altered using an evening-phased promoter (Noordally et al. Science 2013), and in the field (Cano-Ramirez et al. bioRxiv 2022). This phase difference has a several potential explanations. A straightforward interpretation is that the process of *SIG5* translation, chloroplast import and PEP holoenzyme assembly takes some time, thus causing a phase difference. Although it is also possible that the timing difference also involves clock control of chloroplast protein import, clock control of PEP holoenzyme assembly, or post-translational regulation of *SIG5* activity, we note that changing the phase of *SIG5* expression is sufficient to change the phase of *psbD* BLRP expression (Noordally et al. 2013). Because the cause of this delay is likely multifactorial, we reason that its mechanistic basis is beyond the scope of this study- and probably forms the basis for a number of additional publications.

The reviewer’s comment made us realize that we did not consider the topic of the phase differences in our paper. To address this, we added a new Discussion section (lines 325 – 337) about phase delays and other potential mechanisms. We thank the reviewer for asking us to consider this, because it justifies a number of valuable areas of work for the future.

32Previous studies demonstrate that global plastid transcription activity is arrested in the dark but activated by light (Link, 2003; Puthiyaveetil et al., 2010). It has been proposed that phosphorylated PEP and/or sigma factors tightly bind to promoters to arrest transcription under dark conditions. The HY5/HYH -dependent additional signaling might be involved in the regulation of light-dark oscillation of PEP activity which is required for the transcription of many photosynthesis genes, including psbD. I wonder whether the additional signal might regulate the transcription activity of PEP-dependent photosynthesis genes. Circadian analysis of cold-induced transcription activities of whole plastome-encoded genes (Run-on analysis) or circadian CHIP-seq analysis of PEP might shed light on the molecular mechanisms underlying the HY5/HYH -dependent additional signaling.

Response: We agree with the reviewer about these potential explanations. In response to this comment, and request for further Discussion text on the topic by another reviewer, we added an additional paragraph to the Discussion that considers the potential explanations for the modulation of PEP-mediated transcription by light conditions (lines 311-324). A circadian SIG5/PEP ChIP-seq or run-on experiment may well prove informative as part of a future study into circadian regulation of transcription, but it falls outside the scope of our present study because it is not necessary for the conclusions that we reach. We thank the reviewer for these suggestions.

Since phospho-regulation of chloroplast transcription is regulated by chloroplast localized protein kinases, including cpK2, I suppose that it would also be helpful to examine the role of protein kinases in the HY5/HYH-dependent additional signaling.

Response: We agree with the reviewer that a variety of protein kinases could be involved with this process. This is a rather open-ended point with many potential lines of experimentation. We feel that we have addressed this by commenting on potential roles for CSK in the Discussion ("...and the activity of protein kinases thought to modulate sigma factor function such as redox-responsive CHLOROPLAST SENSOR KINASE (CSK) [53, 54]..." lines 334-335).

Interestingly, this study revealed that SIG5 is required for the cold-responsive expression of some nuclear-encoded genes. As suggested by the authors, SIG5 may indirectly regulate the transcription of nuclear-encoded genes, probably via retrograde signals. Please examine the circadian gating of cold-induction of SIG5-dependent nuclear genes in some retrograde signal mutants to identify the retrograde signaling involved in the SIG5-dependent modulation of nuclear-encoded genes.

Response: The nature of any retrograde signals that require SIG5 is a very interesting topic. The reviewer will be aware that there are enormous numbers of potential retrograde signals, ranging from chlorophyll precursors to primary metabolites, mobile RNAs and redox-related processes. The search for potential retrograde signals is a large and open-ended task, potentially representing decades of work. The identity of these retrograde signals is not the focus of this paper and we wish to avoid speculation about their identity. We hope that the text on lines 365-367 encourages other groups to investigate this topic.

This study revealed that SIG5 is required to maintain photosynthetic efficiency and PSII D2 protein levels under long-term exposure to low-temperature conditions and freezing exposure. However, the reviewer felt this study is inconclusive since the response of SIG5 and psbD BLRP transcripts under long-term exposure to low-temperature conditions and freezing exposure is not examined. Photosynthetic efficiency decreases gradually under long-term exposure to low-temperature conditions in WT plants. Is the abundance of PSII D2 protein also decreased gradually under low-temperature conditions? Furthermore, the authors did not examine the response of SIG5 and psbD BLRP transcripts to freezing. Please include these data to examine the possibility that Sig5-dependent maintaining of psbD BLRP transcription is responsible for maintaining the PSII D2 protein level and

photosynthetic efficiency of PSII during prolonged cold and short-term freezing.

Response (first question): The reviewer asked whether PSII D2 protein levels decrease gradually under low temperature conditions. It will be difficult to assign the change in Fv/Fm specifically to a slow change in PSII D2, because *sig5* mutants also affect the abundance of other chloroplast proteins (e.g. PSI-C in Fig. 4), and SIG5 binds several chloroplast operons (Noordally et al. 2013). For this reason, we think that a long search for which proteins change in abundance, at which rates, would be uninformative. This is why we prefer to use Fv/Fm as a read-out of the effect of the *sig5* mutant across several timescales. The

experiment suggested by the reviewer would not change the existing conclusion that *SIG5* maintains photosynthetic efficiency under long-term cold conditions.

Response (second question): We agree that it is interesting to know whether *SIG5* and *psbD* BLRP transcript abundance changes after freezing. Therefore, we conducted the recommended experiment (Fig. S7A), which found that freezing reduces the abundance of both transcripts relative to control temperature conditions. This means that the upregulation of *SIG5* and *psbD* BLRP that occurs under low non-freezing temperatures is not present during freezing, and instead either the presence of *SIG5* or its upregulation at low non-freezing temperatures- before freezing- could maintain photosynthetic capacity during freezing. We describe these new data in the Results ("*SIG5* and *psbD* BLRP transcript levels were decreased relative to control temperature conditions after freezing (Fig. S7A), suggesting the presence of *SIG5* rather than its cold-induction maintains PSII D2 and PSI-C protein abundance during freezing." (lines 275-278). We thank the reviewer for this good suggestion.

Minor comments

Please include the LT-induced expression data of *HY5* and *HYH* transcripts under the experimental conditions used in this study.

Response: Reviewer 3 was interested to know what happens to *HY5* and *HYH* transcripts under our experimental conditions. We agree this is helpful, so collected these data and summarize the finding concisely ("*HY5* and *HYH* transcript levels were upregulated by a 3 h cold treatment at either ZT25 or ZT37 (Fig. S3D), with the exception of *HYH* in the *Ws* background at ZT37." Lines 159-161).

p5 line 108, *psbD* BLRP was arrhythmic in *hyh* (Fig. S3B). correct?

Response: Reviewer 3 asked us to double check whether *psbD* BLRP is really arrhythmic in the *hyh* mutant (Fig. S3B). We understand the reason for this question, given the pattern of the data. We double-checked the time-series analysis (Dataset S1). Analysis of these data with MetaCycle does not identify a statistically significant rhythm from any of the three algorithms used, or the weighted combination of the algorithms. We think that variation in the data (error bars sometimes overlap between peaks and troughs of the rhythm) mean that the transcript is not called as rhythmic. We prefer to adhere consistently to statistical thresholds, so retained the statement that it is arrhythmic. However, other readers might share the reviewer's question, so we revised the Results to clarify the statistical threshold ("*MetaCycle* BH.Q $p = 0.27$ for *hyh*," line 154-155). We thank the reviewer for noticing this.

p6 line146-p7 line 161, Some numbers of genes were not the same as those shown in the figures. e.g., p7 line 155, "959 transcripts that responded to cold at ZT29". However, the number of cold-responsive transcripts in *Col-0* is 954 in Fig.3 and FigS4. Please check.

Response: We appreciate the reviewers' attention to detail. In our previous submission, there was some data reanalysis during manuscript preparation that might have introduced

this error. We corrected this (line 202). We also double-checked the details throughout this section, and believe they are now consistent and accurate.

p11 line 263-270, I suppose that the AAAGTAAG sequence (AAG-box) in the psbD LRP is not a SIG5-binding site. Previous analyses demonstrated that AAG-binding factor (AGF) specifically binds to the AAG-box and interacts with PEP to promote transcription from the BLRP. Please edit the discussion.

Response: We thank the reviewer for noticing this. In response to this, and also a comment from another reviewer, we removed this section because it was too speculative.

Additional comments;

The abstract is written concisely and provides all necessary information. All figure legends properly provide the statistical processing and sample numbers. This manuscript also provides proper citations.

Response: We thank the reviewer for checking these parts of the paper and welcome their view that it is accurate.

Decision Letter, first revision:

Message: 25th January 2023

Dear Dr. Dodd,

Thank you for submitting your revised manuscript "Low temperature and circadian signals converge upon the sigma factor SIG5, which confers photosynthetic resilience to long-term cold" (NPLANTS-220312614B). It has now been seen by the original referees and their comments are below. The reviewers find that the paper has improved in revision, and therefore we'll be happy in principle to publish it in Nature Plants, pending minor revisions to satisfy the referees' final requests and to comply with our editorial and formatting guidelines. Note that referee #1 has suggested some additional experimentation to dispel any doubts about the specificity of the SIG5 effect.

Thank you again for your interest in Nature Plants Please do not hesitate to contact me if you have any questions.

Sincerely,

[redacted]

Reviewer #1 (Comments for the Author):

The manuscript "Low temperature and circadian signals ..." has strongly improved. There is just one point I wish to address again: There is still some doubt on the specificity of the SIG5-effect possible. It would be easier to accept the conclusion if the protein samples (Fig. 4C and 4E) were also standardized on the chlorophyll content and transcript abundance regulation also shown for a not-SIG5-regulated plastid gene.

Reviewer #2 (Comments for the Author):

This is the second round of review of the manuscript. It is much improved and I have no more further questions although it seems that the regulation of SIG5 by HY5/HYH is not direct.

Two comments:

- 1, The reviewer understands that the title has been changed accordingly, but the current title is too long from my perspective.
- 2, For the conclusion section, it looks like not to draw a conclusion but to discuss the significance of the paper.

Reviewer #3 (Comments for the Author):

The manuscript by Cano-Ramirez, Panter et al. entitled "Low temperature and circadian signals coverage upon the sigma factor SIG5, which confers photosynthetic resilience to long-term cold" has been revised appropriately following the reviewers' comments. The authors included several new data and comments that reinforce the conclusion. The revised version of the manuscript is improved over the original. However, I'd like to ask the authors to address the following issues.

- 1 In the main texts (Results and Methods), the authors describe that both SIG5 and psbD BLRP transcripts accumulation was analyzed after 3h cold treatment". However, figure legends of Fig.1 and Fig.2, phrase "abundance of SIG5 and psbD BLRP transcripts after 3 h (SIG5) and 5 h (psbD BLRP) at 4C". Which is correct? If the latter is correct, please explain why a different cold treatment was used for psbD BLRP.

Author Rebuttal, first revision:

The manuscript "Low temperature and circadian signals ..." has strongly improved. There is just one point I wish to address again: There is still some doubt on the specificity of the SIG5-effect possible. It would be easier to accept the conclusion if the protein samples (Fig. 4C and 4E) were also standardized on the chlorophyll content and transcript abundance regulation also shown for a not-SIG5-regulated plastid gene.

Response: We are pleased that the manuscript has improved substantially.

Response (protein levels): On normalization of protein levels in Fig. 4C and 4E, we have a straightforward approach in-hand to standardize protein levels. RbcL is chloroplast-encoded, not part of the photosystems, and is not thought to be regulated by SIG5 (Noordally et al. 2013). Therefore, we calculated the ratio of PSII D2:RbcL and PSAC::RbcL, and added these as Fig. S6C, D. RbcL levels change little in the *sig5* mutant, suggesting that the *sig5*

mutant did not cause systematic changes in chloroplast protein abundance. We explained this briefly in the Results, “Normalization of the abundance of PSII D2 and PSAC protein abundance to the abundance of RbcL under each treatment confirms the reduced abundance of these photosystem proteins relative to RbcL (Fig. S6C, D).” (lines 279-281) and revised the previous sentence to make clear that RbcL is not regulated by SIG5). We also noticed inconsistency in naming of PSAC (PsaC, PSI-C), so we standardized to PSAC according to the information on TAIR. We appreciate the suggestion of normalization to chlorophyll content, but think that approach is unwise because of the risk of independent effects of the combination of light and cold temperatures upon chlorophyll content.

Response (transcript levels): As we explained previously- and supported by the transcriptomic data included in this manuscript- there is not a systematic change in chloroplast transcript levels in the *sig5* mutant. The transcriptional effect of SIG5 upon the blue-light responsive promoter of *psbD* is very specific (Nagashima et al. 2004).

Reviewer #2

This is the second round of review of the manuscript. It is much improved and I have no more further questions although it seems that the regulation of SIG5 by HY5/HYH is not direct.

Response: We are pleased to hear that our manuscript is better.

Two comments:

1, The reviewer understands that the title has been changed accordingly, but the current title is too long from my perspective.

Response: We agree that the title was too long. We revised it to, “Low temperature and circadian signals are integrated by the sigma factor SIG5.” We think this balances title length and specificity, with other details of the study provided in the abstract.

2, For the conclusion section, it looks like not to draw a conclusion but to discuss the significance of the paper.

Response: We feel the end of the paper is a good place to make some general comments about how the findings extend beyond flowering plants. In response to the reviewer’s comment, we revised the section subheading to “Conclusions and perspectives,” to describe the section accurately (line 369).

The manuscript by Cano-Ramirez, Panter et al. entitled "Low temperature and circadian signals coverage upon the sigma factor SIG5, which confers photosynthetic resilience to long-term cold" has been revised appropriately following the reviewers' comments. The authors included several new data and comments that reinforce the conclusion. The revised version of the manuscript is improved over the original. However, I'd like to ask the authors to address the following issues.

1 In the main texts (Results and Methods), the authors describe that both SIG5 and psbD BLRP transcripts accumulation was analyzed after 3h cold treatment". However, figure legends of Fig.1 and Fig.2, phrase "abundance of SIG5 and psbD BLRP transcripts after 3 h (SIG5) and 5 h (psbD BLRP) at 4C". Which is correct? If the latter is correct, please explain why a different cold treatment was used for psbD BLRP.

Response: We apologise that the duration of cold treatments was not explained clearly. The reason for measuring *psbD* BLRP transcripts after a longer cold treatment is straightforward: this plastid transcript takes longer to respond to cold than nuclear-encoded *SIG5*. This might relate to the time taken by the signal to reach the chloroplast (e.g. time for *SIG5* protein import).

We revised the Methods to explain this, "...SIG5 and psbD BLRP transcript abundance was measured after 3 h and 5 h cold treatment respectively, because there is a time delay between accumulation of SIG5 transcripts and downstream psbD BLRP [24, 26]. For gating experiments in the *hy5*, *hyh* and *hy5 hyh* backgrounds, both SIG5 and psbD BLRP abundance was measured at the same time point (after 3 h of cold treatment)." (lines 397- 401). We also double-checked the Results to ensure that the cold treatment durations were explained accurately. In two cases, the cold treatment for *psbD* BLRP measurement was described incorrectly as 3 h (lines 110, 116), which we corrected to 5 h. Finally, for further clarity, we revised the legend of Fig. 1 to add a brief explanation ("*SIG5* and *psbD* BLRP transcript abundance was measured after 3 h and 5 h cold treatment, respectively, because there is a time delay between accumulation of *SIG5* transcripts and downstream *psbD* BLRP [24, 26]." Lines 603-604). We thank Reviewer 3 for asking us to add specific details.

Final Decision Letter:

Message 20th February 2023

:
Dear Professor Dodd,

We are pleased to inform you that your Article entitled "Low temperature and circadian signals are integrated by the sigma factor SIG5", has now been accepted for publication in

1Nature Plants.

Over the next few weeks, your paper will be copyedited to ensure that it conforms to Nature Plants style. We look particularly carefully at the titles of all papers to ensure that they are relatively brief and understandable.

Once your paper is typeset, you will receive an email with a link to choose the appropriate publishing options for your paper and our Author Services team will be in touch regarding any additional information that may be required.

You will not receive your proofs until the publishing agreement has been received through our system

Acceptance of your manuscript is conditional on all authors' agreement with our publication policies (see <http://www.nature.com/authors/policies/index.html>). In particular your manuscript must not be published elsewhere and there must be no announcement of the work to any media outlet until the publication date (the day on which it is uploaded onto our website).

Please note that *Nature Plants* is a Transformative Journal (TJ). Authors may publish their research with us through the traditional subscription access route or make their paper immediately open access through payment of an article-processing charge (APC). Authors will not be required to make a final decision about access to their article until it has been accepted. [Find out more about Transformative Journals](https://www.springernature.com/gp/open-research/transformative-journals)

Authors may need to take specific actions to achieve [compliance](https://www.springernature.com/gp/open-research/funding/policy-compliance-faqs) with funder and institutional open access mandates. If your research is supported by a funder that requires immediate open access (e.g. according to [Plan S principles](https://www.springernature.com/gp/open-research/plan-s-compliance)) then you should select the gold OA route, and we will direct you to the compliant route where possible. For authors selecting the subscription publication route, the journal's standard licensing terms will need to be accepted, including self-archiving policies. Those licensing terms will supersede any other terms that the author or any third party may assert apply to any version of the manuscript.

2An online order form for reprints of your paper is available at <https://www.nature.com/reprints/author-reprints.html>. All co-authors, authors' institutions and authors' funding agencies can order reprints using the form appropriate to their geographical region.

We welcome the submission of potential cover material (including a short caption of around 40 words) related to your manuscript; suggestions should be sent to Nature Plants as electronic files (the image should be 300 dpi at 210 x 297 mm in either TIFF or JPEG format). Please note that such pictures should be selected more for their aesthetic appeal than for their scientific content, and that colour images work better than black and white or grayscale images. Please do not try to design a cover with the Nature Plants logo etc., and please do not submit composites of images related to your work. I am sure you will understand that we cannot make any promise as to whether any of your suggestions might be selected for the cover of the journal.

With kind regards,

[redacted]